
# Tropospheric water vapor profiles obtained with FTIR: comparison with balloon-borne frost point hygrometers and influence on trace gas retrievals

Ivan Ortega[1], Rebecca Buchholz[1], Emrys Hall[2,3], Dale Hurst[2,3], Allen Jordan[2,3], and James W. Hannigan[1]

[1]Atmospheric Chemistry Observations & Modeling, National Center for Atmospheric Research, Boulder, Colorado, USA
[2]Cooperative Institute for Research in Environmental Sciences, University of Colorado, Boulder, Colorado, USA
[3]NOAA Earth System Research Laboratory, Global Monitoring Division, Boulder, Colorado, USA

*Correspondence to:* Ivan Ortega (iortega@ucar.edu); James W. Hannigan (jamesw@ucar.edu)

**Abstract.** Retrievals of vertical profiles of key atmospheric gases provide a critical long-term data record from ground-based Fourier Transform InfraRed (FTIR) solar absorption measurements. However, the characterization of the retrieved vertical profile structure can be difficult to validate, especially for gases with large vertical gradients and spatial-temporal variability such as water vapor. In this work, we evaluate the accuracy of the most common water vapor isotope ($H_2^{16}O$, hereafter WV)

FTIR retrievals in the lower and upper troposphere - lower stratosphere. Coincident high-quality vertically resolved WV profile measurements obtained from 2010 to 2016 with balloon-borne NOAA Frost Point Hygrometers (FPH) are used as reference to evaluate the performance of the retrieved profiles at two sites: Boulder, Colorado and in the mountain top observatory of Mauna Loa, Hawaii. For a meaningful comparison, the spatial-temporal variability has been investigated. Additionally, we evaluate the quantitative impact of different a priori profiles in the retrieval of WV vertical profiles using un-smoothed comparisons. An

orthogonal linear regression analysis shows the best correlation among all layers using ERA-Interim (ERA-I) a priori profiles. In Boulder, we found a negative bias of $0.02 \pm 1.9$ % and precision of 3.7 % (r = 0.95) for the 1.5 - 3 km layer. A larger negative bias of $11.1 \pm 3.5$ % and precision of 7.0 % was found in the lower free troposphere layer of 3 - 5 km (r = 0.97) attributed to rapid vertical change of WV, which is not always captured by the retrievals. The bias improves in the 5 - 7.5 km layer ($1.0 \pm 5.3$ %) and the precision worsens to about 10 %. The bias remains at about 13 % and the precision remains to about 10 % for

layers above 7.5 km but below 13.5 km. At MLO the spatial mismatch is significantly larger due to the launch of the sonde being farther from the FTIR location. Nevertheless, we estimate a negative biases of $5.9 \pm 4.6$ % for the 3.5 - 5.5 km layer (r = 0.93) and $9.9 \pm 3.7$ % for the 5.5 - 7.5 km layer (r = 0.93), and positive biases of $6.2 \pm 3.6$ % for the 7.5 - 10 km layer (r = 0.95), and 12.6 % and greater values above 10 km. The agreement for the first layer is significantly better at BLD likely that the air masses are similar for both FTIR and FPH. Furthermore, for the first time we study the influence of different sources of

WV profiles in the retrieval of selected gas profiles. Using NDACC standard retrievals we present results for hydrogen cyanide (HCN), carbon monoxide (CO), and ethane ($C_2H_6$) by taking NOAA FPH profiles as the ground-truth and evaluate the impact of other WV profile sources. We show that the effect is minor for $C_2H_6$ (bias < 0.5 % for all WV sources) among all vertical layers. However, for HCN we found significant biases between 6 % for layers close to the surface to 2 % for upper troposphere





depending on WV profile source. The best results (lowest bias/precision and r-values closer to unity) are always found for pre-retrieved WV. Therefore, we recommend to first retrieve WV to use in subsequent retrieval of gases.

# 1 Introduction

Water vapor is an ubiquitous atmospheric constituent with an extremely important role in the lower and middle troposphere and
stratosphere: it is the most variable and critical greenhouse gas (Kiehl and Trenberth, 1997); it plays a key role in atmospheric chemistry, e.g., heterogeneous chemistry, aerosol formation, and wet deposition (Seinfeld and Pandis, 2006); it affects global radiation through cloud formation (Dessler, 2011); and acts as the main source for precipitation in the lower atmosphere (Trenberth and Asrar, 2014). Middle and upper tropospheric and lower stratosphere stable water vapor isotopes are key to understanding the water cycle feedbacks such as mixing of air masses, dehydration pathways, and free-tropospheric moisture
(Noone, 2012; Galewsky and Rabanus, 2016).

Obtaining consistent long-term observations of vertical distributions of water vapor is challenging but highly desirable in order to understand climate evolution and feedback effects (Held and Soden, 2000). There is a need to measure water vapor vertical distribution for long-term monitoring but there are only few data sets, e.g., in-situ balloon observations in Boulder, Colorado, USA are the longest data set of the most common water vapor isotope ($H_2^{16}O$, hereafter $H_2O$ or WV) with informa-
tion from lower to middle stratosphere (Oltmans et al., 2000; Hurst et al., 2011b). It has been shown that ground-based Fourier transform infrared (FTIR) measurements provide reliable long-term and continuous observations of WV (Sussmann et al., 2009; Schneider et al., 2010). FTIR measurements have focused mostly on integrated WV analysis among the Network for Detection of Atmospheric Composition Change (NDACC, see http://ndacc.org). For integrated WV (IWV, i.e., total columns) FTIR have been shown to be very precise with about 2.2 % using FTIR side-by-side inter-comparisons (Sussmann et al., 2009).
The retrieval of WV vertical profiles with degrees of freedom (DOF) larger than two are achieved but there is a lack of comprehensive comparisons of vertical gradients using FTIR with well-characterized highly-resolved independent measurements from the lower to upper troposphere and stratosphere. Comparisons of FTIR and operational radiosondes have been used to validate optimized retrieval strategies, e.g., for WV in the upper troposphere/lower stratosphere see Schneider et al. (2006) and Schneider and Hase (2009) for tropospheric approaches. The MUSICA project (MUlti-platform remote Sensing of
Isotopologues for investigating the Cycle of Atmospheric water) within the scope of NDACC aims to characterize long-term observations of the ratio of water vapor isotopologues in the lower/middle troposphere (Schneider et al., 2012, 2016; Barthlott et al., 2017). Vogelmann et al. (2015) studied the spatial-temporal variability of WV in the free troposphere (Zugspitze,, Germany) by exploiting the geometry of measurements of differential absorption lidar (DIAL) and FTIR. In particular, they assessed the variability in short scales, i.e., few kilometers and minutes.
In this work, we evaluate the accuracy and precision of WV profiles using a standard retrieval inversion with ground based FTIR measurements. For the first time, the retrieval validation uses coincident and well-characterized balloon-borne in-situ NOAA frost point hygrometer (FPH) measurements (Hall et al., 2016). The FPH measurement technique has been used as reference to assess the accuracy of radiosonde relative humidity measurements due to their high vertical time resolution and



low uncertainties (Suortti et al., 2008; Hurst et al., 2011a). Understanding the significance of the WV a priori profiles is important for WV due to its high temporal and spatial variability. Especially, for un-smoothed comparisons with the goal to assess WV vertical gradients. In this work, we assess the influence of the different WV a priori profiles in the retrieval of WV at several altitudes. Finally, it is well-known that the strong WV absorption signatures interfere in the retrieval of other gases.

However, there is a lack of quantitative effect of WV at different altitudes. This study also provides a quantitative assessment of the impacts of WV in the retrieval of selected tropospheric gases, hydrogen cyanide (HCN), carbon monoxide (CO), and ethane ($C_2H_6$), using NDACC standard retrievals.

## 2   Measurements

### 2.1   Free tropospheric and boundary layer FTIR sites

FTIR direct solar IR absorption spectra are measured under clear-sky conditions in two different locations: (1) Boulder, Colorado (hereafter BLD; 40.40° N, 105.24° W, 1600 m.a.s.l) and (2) Mauna Loa, Hawaii (hereafter MLO; 19.40° N, 155.57° W, 3400 m.a.s.l). The spectra at BLD have been recorded using a Bruker 120 HR spectrometer operated since 2010 following standard measurement protocols of the Infra Red Working Group (IRWG)/NDACC (http://ndacc.org). The instrument is located in the foothills laboratory of the National Center for Atmospheric Research (NCAR) situated in the front range of the

Rocky Mountains and within the planetary boundary layer. Previous studies have used the BLD dataset for satellite validation of $NH_3$ (Dammers et al., 2017), mobile low resolution FTIR validation of $NH_3$ and $C_2H_6$ (Kille et al., 2017); and analysis of gases emitted by oil and natural gas development (Franco et al., 2016; Tzompa-Sosa et al., 2016). The MLO instrument has been part of the long-term activities of the IRWG/NDACC. First IR solar absorption spectra were recorded at MLO in 1991 using a Bomem DA02. In 1995 a Bruker 120 M started to operate and was upgraded in 2011 to a Bruker 125 HR. The high

altitude site at MLO is normally above the boundary layer and the measurements are sensitive mainly to free tropospheric and stratospheric air masses. At both sites the spectra are recorded using optical band pass filters maximizing the signal to noise ratio (SNR) over the near and mid-infra red spectral domain with a nominal spectral resolution of 0.0035 cm$^{-1}$ using liquid nitrogen-coolded InSb and MCT detectors and a KBr beam-splitter (Hannigan et al., 2009).

### 2.2   Balloon-borne NOAA Frost Point Hygrometer

Highly precise and accurate in situ measurements of tropospheric and stratospheric WV over Boulder, Colorado, and Hilo, Hawaii, are performed with balloon-borne FPHs by the Global Monitoring Division of NOAA's Earth System Research Laboratory (ESRL). These measurements are also part of the GCOS Reference Upper Air Network (GRUAN) and the NDACC. At both sites, balloon-borne FPHs are launched once per month, preferably during conditions of low winds and clear skies. The Boulder measurements started in 1980 and are launched at Marshall Field Site (1743 m.a.s.l), 10.5 km south of the BLD

FTIR measurement site (Oltmans et al., 2000; Scherer et al., 2008; Hurst et al., 2011b). Monthly NOAA FPH soundings at Hilo started in 2010 and the balloons are launched from the National Weather Service facility at Hilo International Airport (10





m.a.s.l), 58.0 km east of MLO. In this paper we emphasize the comparisons at BLD due to the shorter distance between the FTIR and balloon launch site, although we perform identical comparisons and present results from MLO as well.

A thorough description of the FPH measurement technique has been described elsewhere (Hurst et al., 2011b; Hall et al., 2016). Briefly, the basic principle is to condense WV from a stream of air onto a small, gold-plated mirror using a cryogenic liquid to continually cool the mirror. Once a thin condensed layer is deposited on the mirror, pulses of heat are applied as needed to maintain a stable layer of condensate. Changes in frost (ice) coverage are detected by measuring the mirror reflectivity using a small LED-based infrared beam and a photodiode. The amount of heat applied is rapidly adjusting to produce a stable frost layer, at which point the temperature of the mirror (frost point temperature) is a direct measure of the partial pressure of WV in the air stream above it via the Goff-Gratch equation (Goff, 1957). The water vapor mixing ratio is calculated by dividing the WV partial pressure by the dry atmospheric pressure. Since a FPH fundamentally makes temperature measurements, only the thermistor embedded in each mirror requires calibration. Each thermistor is calibrated using NIST traceable standards (see Hall et al. (2016)). A recent detailed analysis of WV mixing ratios measured by the NOAA FPH shows the uncertainties (2-sigma) are < 12 % for the 0 - 5 km altitude layer, < 8 % for 5 - 13 km, and < 6 % for 13 - 28 km (Hall et al., 2016). The NOAA FPH vertical profile data employed here are 0.25 km vertical averages and their standard deviations that are calculated from the measurements made at 5-10 m vertical resolution during balloon ascent.

## 3   Retrieval of water vapor from FTIR

Prior to the retrieval of WV from the solar absorption spectra a quality control of each measurement is carried out, i.e., visual inspection of spectra and assessment of the SNR. As mentioned in Sect. 2.1, we only use spectra taken during cloud-free conditions. The spectra are analyzed using the retrieval code SFIT4 v4.0.9.4, which has been improved from its predecessor SFIT2 (Pougatchev et al., 1995; Rinsland et al., 1998; Hase et al., 2004). SFIT4 derives vertical profiles and the corresponding density vertical columns from the pressure broadening and temperature dependency of specific absorption lines. The overall retrieval follows the optimal estimation method applied to several micro-windows. The inverse problem is ill-posed and the solution is constrained by an a priori profile ($x_a$) and covariance matrix ($\mathbf{S_a}$), which ideally should represent the natural variability of the WV profile from climatological records (Rodgers, 2000; Rodgers and Connor, 2003). Section 4.3 describes in more detail the effect of the a priori and the different a priori profiles used in this study. In many cases $\mathbf{S_a}$ is not well-known due to lack of long-term highly-resolved profiles and ad hoc constraint are used (e.g., Vigouroux et al. (2015)). The constrained solution is important in order to avoid many different atmospheric states in the minimization of the cost function following a Gauss-Newton iteration:

$$x_r = x_a + \mathbf{S}_a \mathbf{K}^T \left( \mathbf{K} \mathbf{S}_a \mathbf{K}^T + \mathbf{S}_e \right) \left( y - \mathbf{K} x_a \right) \tag{1}$$

where $x_r$ is the retrieved state vector, $\mathbf{K}$ is the weighting function, $\mathbf{S}_e$ is the measurement noise covariance matrix, and $y$ is the measurement state vector.

Many of the spectral windows used to retrieve NDACC gases contain WV absorption signatures. Retrieved WV is often used in the retrieval of other gases, because accurate quantification of the interfering WV reduces retrieval uncertainty. WV can be



**Table 1.** Micro-windows for $H_2O$ retrieval including interfering gases retrieved within those micro-windows. Column gases are those retrieved by profile scaling of initial profile while profile retrieval is done for the profile gases column.

| Micro-window [$cm^{-1}$] | Profile Gase(s) | Column Gase(s) |
|---|---|---|
| 1) 2611.40 - 2613.40 | HDO | $CO_2$ |
| 2) 2659.00 - 2661.00 | HDO, $CH_4$ | $CO_2$ |
| 3) 2819.00 - 2819.80 | $H_2O$, $CH_4$ | $N_2O$, HCl |
| 4) 2829.80 - 2839.40 | $H_2O$, $CH_4$, HDO | - |

retrieved using multiple micro-windows since it absorbs from the near to far infrared wavelengths. With the goal to characterize this 'pre-retrieved' WV we use retrieval settings that are commonly used among NDACC sites. We use the 2600 - 2840 $cm^{-1}$ spectral domain to simultaneously retrieve $H_2O$ and the isotopolog HDO. In this study, we focus only on $H_2O$. A short summary of the four micro-windows and interfering species included in the analysis is given in table 1. The spectroscopic data used here

is based on the line-by-line portion of the HITRAN 2008 (Rothman et al., 2013). The errors in the reported line parameters are described in section 3.1 and are used to estimate the systematic uncertainty in the retrieval. Most of the interfering species are fit as a scaling of the a priori vertical profile ($CO_2$, $N_2O$ and HCl) with the exception of $CH_4$ which is fit as a profile in micro-window two, three, and four. The SNR determines how much influence the spectra has in each micro-window versus the a priori, as well as to characterize the measurement error described in section 3.1. In order to prevent sporadic vertical profile

oscillations due to relaxed covariance matrices we implement ad hoc diagonal elements of $\mathbf{S_a}$ with a maximum variability of 50% at the surface and exponentially decreasing by altitude with a inter-layer thickness correlation coefficient. A Gaussian correlation with a length of 25 km is used for the off-diagonal elements of $\mathbf{S_a}$. The instrumental line shape (ILS) has been fixed with a unity modulation efficiency and ideal phase error. Schneider et al. (2012) pointed out that the ILS does not play an important role in the WV error budget and is of lower importance for tropospheric WV retrievals.

Inputs into SFIT4 include vertical profiles of pressure, temperature, and the volume mixing ratios (VMR) of the atmospheric gases included in the fit. Preceding to the retrieval, SFIT4 employs the Air Mass Computer Program for Atmospheric Transmittance/Radiance Calculation (FSCATM) ray tracing code to calculate the atmospheric path (Hannigan et al., 2009). The input pressure and temperature vertical profiles are obtained from the National Center for Environmental Prediction (NCEP) reanalysis based on the NCEP/NCAR analysis/forecast system to perform data assimilation using past data from 1948 to the present

(Kalnay et al., 1996). These profiles are obtained directly from NDACC (http://ndacc.org). These are daily average profiles that extend to up to 0.4 mb (approximately 50 km). Above 0.4 mb we use monthly mean pressure and temperature profile from an average of a 40 year simulation (1980-2020) of the Whole Atmosphere Community Climate Model (WACCM) (Garcia et al., 2007). These profiles are merged using a cubic spline interpolation for pressure and a quadratic spline interpolation for temperature. We examined the effect of using more temporally refined temperature profiles. In general, the root mean square error

(rmse) of the fit between the six hourly data of ERA-I reanalysis model produced by the European Center for Medium-Range Weather Forecasts (ECMWF) (Dee et al., 2011) and daily temperature is less than 0.5% of the mean temperature using 2013




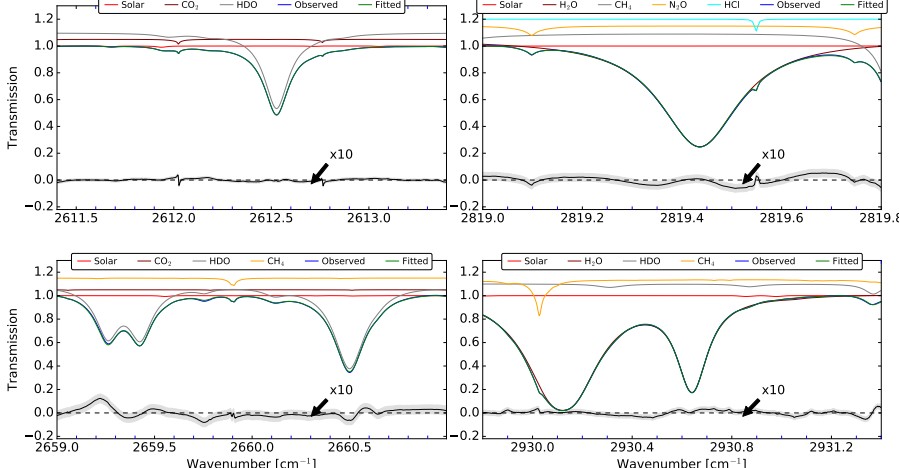

**Figure 1.** Mean retrieval fit between 2010 - 2016 for the spectral intervals of WV. The observed and fitted lines are blue and green respectively. The absorption contribution for the different species are also shown in each micro-window. The bottom black lines represent the mean residual and the gray shadow are the standard deviation. Note that for visibility the residuals have been multiplied by 10.

data and the bias of the fits are insignificant indicating that the daily mean temperature profile is adequate for the retrievals. With the exception of WV (see section 4.3), VMR input mean profiles of all other gases are taken from the 40 year run of WACCM.

### 3.1 Characterization and error budget

The mean retrieval fit of the four micro-windows between 2010-2016 at BLD is shown in Fig. 1. The small systematic residual structures (black lines) are likely caused by spectroscopic parameter error but in general the magnitude of residuals is low and within noise level ($< 0.1\,\%$).

The information content of the retrieved WV vertical profile is characterized by means of the averaging kernel matrix, $\mathbf{A}$:

$$\mathbf{A} = \left(\mathbf{K}^T \mathbf{S}_e^{-1} \mathbf{K} + \mathbf{S}_a^{-1}\right)^{-1} \mathbf{K}^T \mathbf{S}_e^{-1} \mathbf{K} \tag{2}$$

The rows of the mean $\mathbf{A}$ known as averaging kernels (AK) at BLD obtained between 2010 - 2016 color coded by altitude below 20 km are shown in Fig. 2a. The maximum values are located at the surface, then they decrease and remain steady to about 8 km and eventually decrease to zero above 12 km. This indicates that most of the information content obtained from WV pressure dependence of the absorption features is limited to the troposphere. The mean total column averaging kernel (TAK) is shown in Fig. 2b. Tipically, a unity TAK indicates that the retrieval is not biased, while values of the TAK lower than unity indicate underestimation and larger values than unity indicate overestimation of the real WV. Hence, below 3 km the retrieval

may underestimate, between 3-8 km overestimate, and between 8-12 km underestimate the real WV magnitude. The mean number of degrees of freedom (DOF), given by the trace of the $\mathbf{A}$, are 2.4 and indicate the total number of independent pieces





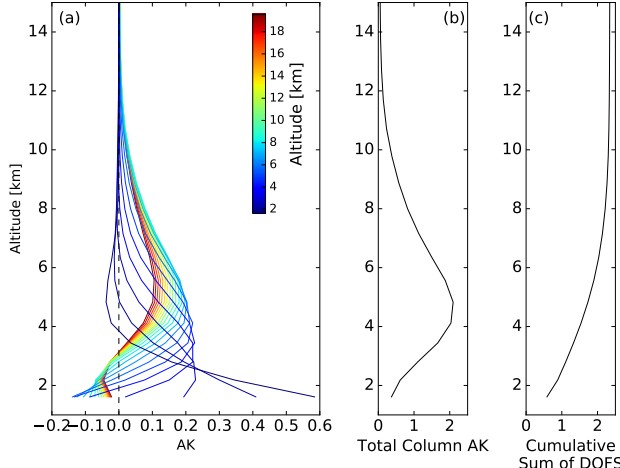

**Figure 2.** (a) FTIR mean row averaging kernels; (b) mean total column averaging kernel; and (c) cumulative sum of DOF of WV obtained in BLD from 2010 - 2016.

of information in the retrieval. The vertical profile of the cumulative sum of DOF is shown in Figure 2c and shows that the first DOF is given in the layers below 3 km, the second DOF is given between 3 - 6 km, and the rest above. Further optimization of the retrieval strategy might improve the $\mathbf{A}$ but as explained before one of the goals is to assess the current retrieval strategy, therefore we do not investigate retrieval constraints further. At MLO the vertical sensitivity is similar but starting at 3.5 km. A

proper comparison between FTIR and in-situ sonde profiles would require to smooth the in-situ measurements using the FTIR AK and a priori to account for lower resolution measurement's smoothing (Rodgers and Connor, 2003). However, as pointed out by Schneider et al. (2006) the information of the WV $\mathbf{A}$ is limited due to its high variability through the troposphere. Additionally, smoothing the in-situ measurements would require using the a priori profile, which in turn may be highly biased to the real atmospheric state. A goal of the present study is to determine whether the highly structural variability of WV

can be retrieved with FTIR measurements, hence the comparison with in-situ sonde measurements are carried out without a smoothing.

    SFIT4 estimates an uncertainty budget that combines random, systematic, and smoothing sources following the formalism given in Rodgers (2000). The most important random error is normally the retrieval noise characterized with the SNR in the spectral region of interest. The error covariance matrix ($\mathbf{S_n}$) is calculated with the following equation:

$$\mathbf{S}_{\mathrm{n}} = \mathbf{G}_{\mathrm{y}} \mathbf{S}_{\mathbf{e}} \mathbf{G}_{\mathrm{y}}^{\mathbf{T}} \tag{3}$$

where the gain matrix $\mathbf{G}_{\mathrm{y}}$ represents the sensitivity of the retrieval to the measurement and is related with the averaging kernel as $\mathbf{A} = \mathbf{G}_{\mathrm{y}}\mathbf{K}$. Currently, the diagonals of the $\mathbf{S_e}$ matrix are constructed using the square of the inverse of the SNR obtained from the noise in the spectra of interest, and off diagonal elements are not considered. The retrieval of WV is actually an estimate of a state smoothed by the averaging kernel. The difference between these two states is given by the smoothing error



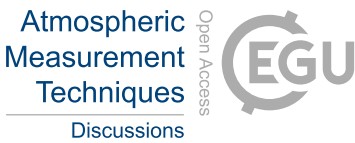

$(\mathbf{S}_s)$:

$$\mathbf{S}_s = (\mathbf{I} - \mathbf{A})\mathbf{S}_a(\mathbf{I} - \mathbf{A})^T \tag{4}$$

The smoothing error is treated separately and not included in the total error analysis because $\mathbf{S}_a$ is normally not well known and consequently is often simplified. The model parameter error represent the errors in the forward model parameters such as

temperature, solar zenith angle (SZA), and spectroscopic parameters. These errors can contain both systematic and random components. We obtain the model parameter covariance matrix as:

$$\mathbf{S}_b = (\mathbf{G}_y\mathbf{K}_b)\mathbf{S}_b(\mathbf{G}_y\mathbf{K}_b)^{\mathbf{T}} \tag{5}$$

Where, $\mathbf{S}_b$ is the error covariance matrix on the model parameters. The largest contributors are considered here and are the absorption line parameters, temperature profiles, and SZA. The uncertainty of the absorption line parameters, i.e., line

intensity ($S$), air-broadened half width ($\gamma$), and temperature dependence of $\gamma$ ($n$), are taken from the lower limit reported in HITRAN 2008 (Rothman et al., 2013). These uncertainties are only considered systematic and the errors reported in HITRAN are 5, 1, and 10 % for $S$, $\gamma$, and $n$, respectively. The error in the temperature profile is considered to have both systematic and random components. These errors have been quantified with the mean (systematic) and standard deviation (random) of the difference of long-term comparisons between NCEP profiles with radiosondes launched near the sites and/or ERA-I reanalysis.

The measurement noise error is estimated with the the square of the inverse of the SNR as diagonal elements in the covariance matrix. The pointing accuracy in the SZA is considered random and has been characterized with an error of 0.15°.

Figure 3 shows the random and systematic vertical profile uncertainties in percentage with respect to the mean mixing ratio. The major systematic components in the lower troposphere are the absorption line parameters $S$ and $\gamma$ but in the upper troposphere the temperatures contributes equally. The temperature and measurement noise are the main components of the

random uncertainty. The final uncertainty is estimated from the error propagation of all components and is lower than 10 % below 4 km and about 10 % above. The instrumental line shape uncertainty plays a minor role in the total error budget.

## 4  Comparison of water vapor vertical profiles

The total number of sonde observations are 90 at Boulder and 70 at Hilo from 2010 to 2016. The overall number of coincident dates of measurements under ideal conditions are 56 and 36 for BLD and MLO, respectively. Figure 4 presents a rough

qualitative comparison of selected WV profiles obtained with NOAA FPH measurements and FTIR retrievals in BLD. The daily mean ERA-I (henceforth ERA-d) a priori profiles used in the retrievals are also shown. To retain high vertical variability the FPH profiles are shown in 0.25 km vertical averages of the sonde's ascent measurements. The FTIR profiles represent the average profile weighted by the error and the blue shading depicts the uncertainties propagated using the individual profiles within 2 h of the FPH launch. The temporal variability and its effect are studied in section 4.1. The retrieved WV profiles

capture the vertical gradients identified with the in-situ NOAA FPH even though the a priori profile may be biased (see for example 2010-09-14 and 2010-11-05). For comparison, Fig. S1 in the Supplement also shows the same figure but smoothing the sonde profiles using the FTIR averaging kernel and a priori to account for lower resolution measurement's smoothing





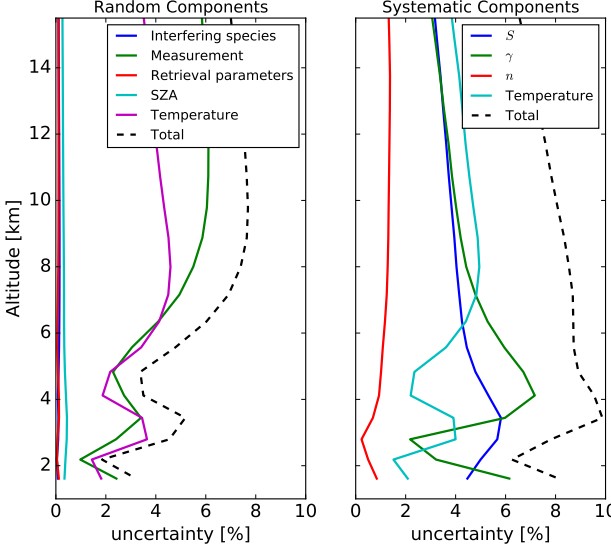

**Figure 3.** Mean vertical profiles of the most important random (left) and systematic (right) uncertainty components for the retrieval of WV in BLD from 2010 - 2016.

(Rodgers and Connors, 2003). However, it is clear that smoothing diminishes the real variability, which is actually captured by the retrieval. Figure 5 shows the same but for selected vertical profiles at MLO. The near-surface mixing ratios at this high-altitude site are significantly lower and the profiles show steeper vertical gradients that at BLD. Note that the FTIR (MLO) and FPH (Hilo) are about 60 km apart and might have sampled different air masses. In BLD the launch of the FPH is only 10 km south of the ground-based FTIR. The generalized comparison using smoothed FPH profiles at MLO are also shown in the Supplement (see Fig. S2).

To quantitatively compare both measurements the high vertical resolution balloon-borne profiles are re-gridded onto the altitude grid of the FTIR retrieval by means of a linear interpolation. For BLD the nearest FPH point to the surface is typically few hundred meters above the first grid point of the FTIR (see Fig. 4). In this case, we assume homogeneous WV close to the surface and use the nearest-neighbor point. Due to the limited number of DOF we do not aim to compare every grid point but to assess several layers maximizing the number of points and characterizing boundary layer, free troposphere, and upper troposphere - lower stratosphere. The following layers have been chosen for BLD: 1.5-3.0, 3.0-5.0, 5.0-7.5, 7.5-10, 10-13, and 13-17 km above sea level (asl) and for MLO: 3-5.5, 5.5-7.5, 7.5-10, 10-13, 13-16, and 16-20 km asl. These layers have been chosen so that they include three standard IRWG FTIR grid points. Comparison of ground-based remote sensing with balloon-borne in-situ measurements is challenging due to spatial-temporal variability. The temporal and spatial variability are characterized in the next two sections followed by the quantitative comparison between FTIR and NOAA FPH.





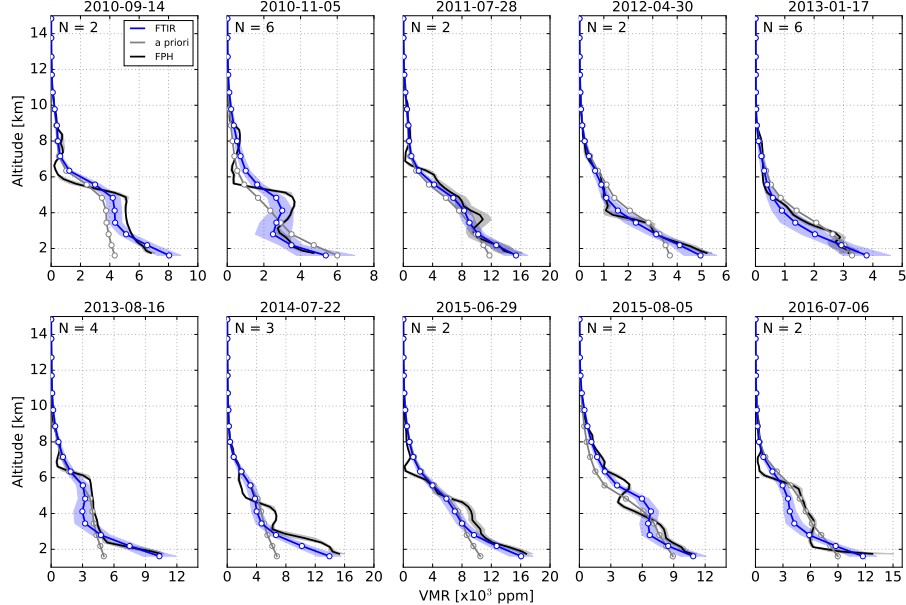

**Figure 4.** Comparisons of WV vertical profiles for selected dates obtained with in-situ NOAA FPH measurements (black) and FTIR retrievals (blue) in BLD. The ERA-d WV used as a priori is shown in gray. The dates are shown at the top of each plot. The FTIR profiles represent weighted mean profiles using retrievals within two hours of the radiosonde launch. The blue filled shadow area represents the standard error propagation using the uncertainty in individual retrievals. The gray shaded are of FPH profiles are the 1-sigma standard deviation of each mixing ratio. The number of retrieved profiles within 2 hours is shown on the upper-left of each panel.

## 4.1 Temporal variability

Due to the lack of independent time-resolved WV vertical profiles we use daily FTIR observations to assess the temporal variability. Figure 6 shows the number of dates and profiles and the variability of WV in percent for several layers as function of the length of time interval starting from 0 to 3 minutes and gradually increasing, e.g., 0 to 10, 0 to 30, 0 to 60 minutes, etc.

5 The retrievals produced during these time intervals are used to calculate the temporal variability using the ratio of the standard deviation to the mean values at several altitude layers. This approach is sensitive only to the variability observed by the FTIR, however the real variability might be greater because of potential lost variability during retrieval smoothing. This proxy for variability has been estimated using dates during coincident measurements between sondes and FTIR. The number of dates and profiles are roughly the same below 10 min, indicating the time that the FTIR takes to start a new measurement using the

10 same band pass filter for a standard set of observations. The number of profiles starts to increase from the number of dates after 15 min. The variability in BLD among different layers does not vary substantially and they remain within 1 - 2 % of each other, indicating similar relative variability within all the different tropospheric and stratospheric layers. In BLD the variability starts to increase from about 1 % in 30 min to 6 % in 240 min. In contrast, at MLO the variability is different among layers. A variability up to 9 % is found for the layer close to the instrument altitude (3 - 5.5 km), however the variability is below 5 % for





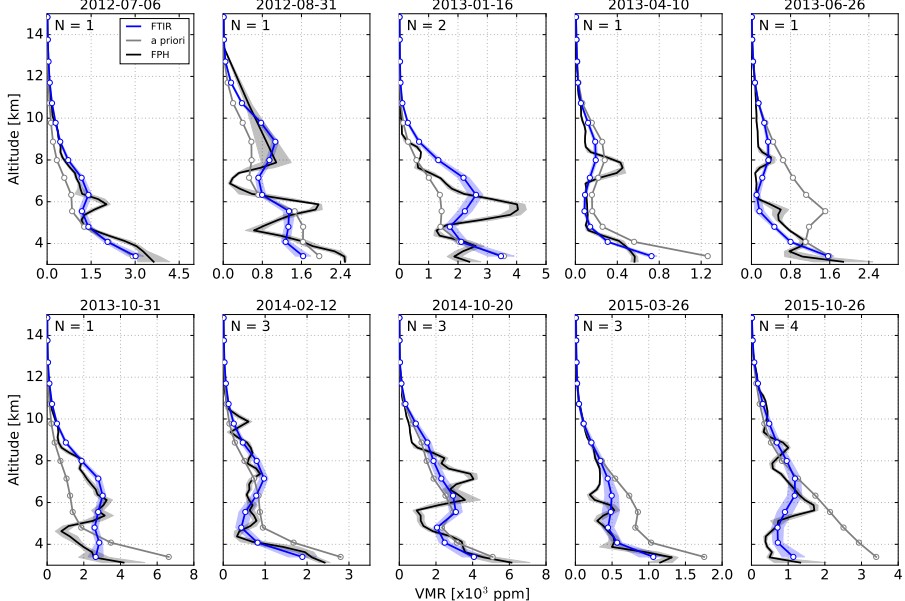

**Figure 5.** Same as Fig. but for MLO.

the layer between 5.5 - 7.5 km, and even below 5 % for the 16 - 20 km indicating a vigorous fluctuations and strong convection near the MLO site. In general, these findings suggests that the coincidence time interval to avoid variability larger than 2 % is 30 min at BLD and 60 min at MLO. The air mass probed by the FTIR is changing during the day due to the line of sight to the sun moving constantly such that after some time the spatial variability may play an important role. Vogelmann et al. (2015) estimated that the spatial mismatch may play a role for intervals longer than 30 min. The spatial mismatch is described in the next section.

## 4.2 Spatial mismatch

If the spatial mismatch between the FTIR and sonde is considerably large they might probe distinctive air masses. Hence, natural WV variability would affect a meaningful comparison (Sussmann et al., 2009; Vogelmann et al., 2015). A thorough assessment of the spatial variability of WV would require measurements of an extensive area simultaneously and at different altitudes. In order to estimate the spatial mismatch we calculate the horizontal distance between the sonde location and the line of sight of the FTIR. The effective horizontal position sensitivity of the FTIR depends on the sun-pointing geometry and the vertical WV profile distribution. We adopted a methodology applied by Vogelmann et al. (2015) to estimate this effective horizontal position. This method assumes that the FTIR sensitivity is located at the point where the viewing direction of the instrument meets the altitude level of the mass weighted WV profile. Using the mass weighted WV of all sonde profiles we estimate roughly an altitude of $3.8 \pm 0.9$ km in BLD. Using this altitude and the SZA the horizontal distance from the ground-based site is calculated for every measurement. Then, using the solar azimuth angle the latitude and longitude are calculated





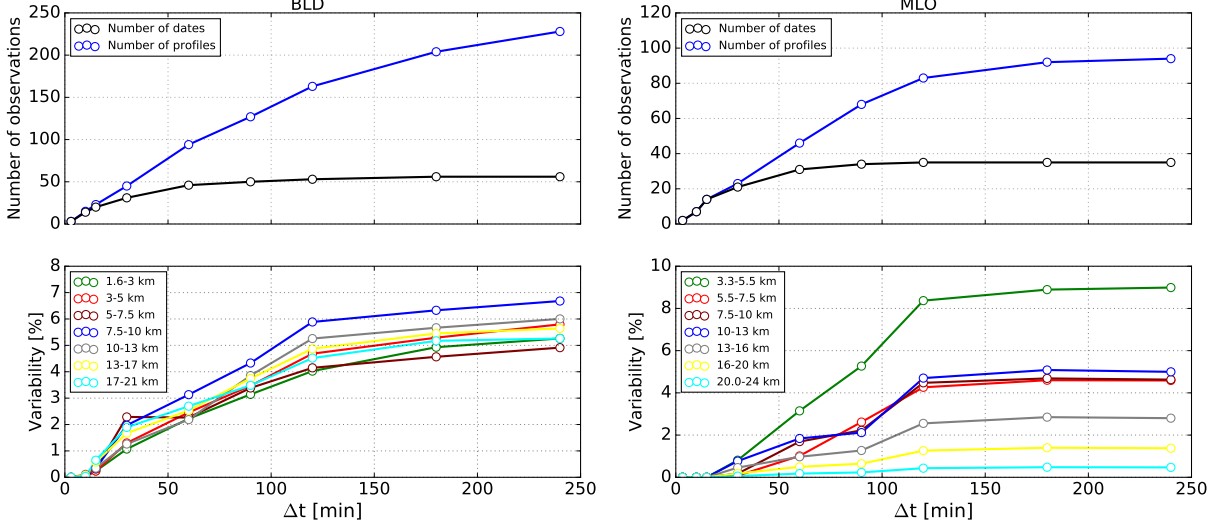

**Figure 6.** The top panels show the number of dates (black) and profiles (blue) measured by the FTIR at BLD (left) and MLO (right) as a function of the length of the time interval in minutes. The bottom panels show the temporal variability in percent estimated with the ratio of the standard deviation to the mean values for several layers as a function of the length of the time interval. The length of the time intervals are defined as increasing temporal window, e.g., 0 - 30, 0 - 60, 0 - 120 minutes, and the number of retrievals in each window is used to calculate the variability.

after having traveled the given distance on the given bearing. Once the location is found the haversine formula is applied to determine the great-circle distance between two locations (Korn and Korn, 2000). At BLD the mean distance with respect to the FTIR site location is $6.0 \pm 4.0$ km south making the initial spatial mismatch with the sonde launch about 6.5 km. At MLO the mass weighted WV profile is $6.0 \pm 0.6$ km and the initial spatial horizontal mismatch is 47.0 km (see Fig. S3 in the

Supplement). Consequently, even co-located sonde launches may not exactly probe the same air mass.

The spatial mismatch at different altitudes depends on the sonde trajectory and the location of the FTIR sensitivity. At BLD the GPS location at every sonde altitude is available for almost all profiles and the horizontal distances between all altitudes and the FTIR sensitivity on the earth are calculated. Figure 7 shows the mean spatial mismatch between the FTIR and the sonde profiles for the coincident time intervals of 0 - 30 and 90 - 120 minutes. As mentioned above the initial spatial difference

close to the surface is about 6 km. For the 0 - 30 min interval the horizontal difference is below 10 km below 4.5 km altitude, similarly for the 90 - 120 min, except for one altitude, which is greater than 15 km. Above 5 km altitude the spatial mismatch starts to increase. A rapid significant increase in the spatial mismatch is identified above 5 km for both 0 - 30 and 90 - 120 min coincident time intervals. Interestingly, the greatest horizontal difference is found for the 0 - 30 min interval with maximum values of about 70 km. This analysis shows that the spatial mismatch depends on the complex convective dynamics and not only

in the coincidence time interval. Nevertheless, only short temporal coincidence differences are encouraged to avoid temporal WV fluctuations as shown above.





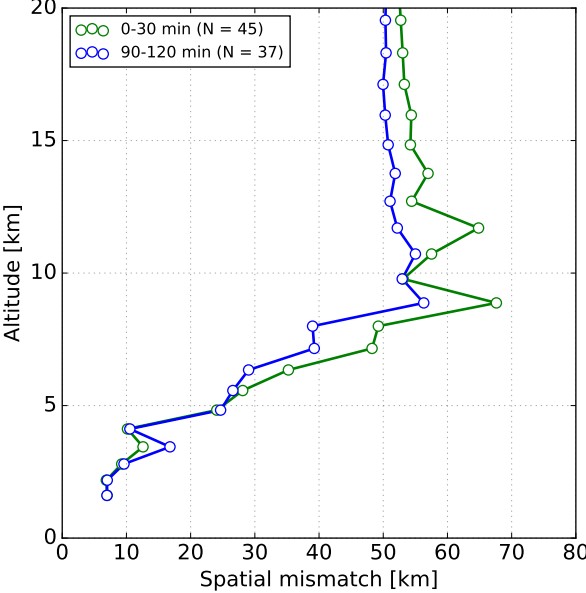

**Figure 7.** Vertical profile of the horizontal spatial mismatch between FTIR and sonde profiles in BLD. As an example two coincident time intervals are used.

### 4.3 Influence of a priori profiles

The a priori profile influences the solution of equation 1 in two ways, one in the first order weighting and also if the retrieval is not in a linear regime (Kulawik et al., 2008). Since WV is highly variable, even in time scale of hours, using the most accurate a priori might improve the retrieval results. Especially, if the comparison are carried out without smoothed in-situ profiles with aim to capture vertical gradients and to avoid limited information in the averaging kernel. Four different a priori profiles are used to retrieve WV, which then are compared with balloon-borne NOAA FPH measurements: (1) a 40 year simulation (1980-2020) of the WACCM monthly mean profiles. WACCM is a global model with 66 vertical levels from the ground to approximately 140 km geometric height, the horizontal resolution is 1.9/2.5° (latitude/longitude) and is part of the NCAR Community Earth System Model (for further details see Garcia et al. (2007); Marsh et al. (2013); Kinnison et al. (2007) ) ; (2) ERA-d; (3) 6 hourly WV vertical profiles (00, 06, 12, and 18 UTC) obtained from ERA-I (ERA-6). In this case, the closest in time to the measurements is used. ERA-I profiles extend to 1 mb and then are merged with WACCM monthly mean profiles of WV using a spline interpolation. We take the closest ERA-I grid point to represent the a priori at each station; and (4) daily NCEP/NCAR (NCEP-d) reanalysis WV profiles (Kalnay et al., 1996). Since the spatial resolution of NCEP is lower (2.5 x 2.5°) we interpolate WV spatially to obtain the best WV profile.

We have chosen the above four a priori profiles since they are readily available and commonly used. An optimization of the data set is carried out before the quantitative comparison. The difference between WV retrievals and sonde profiles ($\Delta x = x_r - x_s$) shows a normal distribution centered around zero for the layers defined in section 4. Fig. S4 in the Supplement shows





an example of the $\Delta x$ distribution using ERA-d for the different layers. Extreme outlieres are identified for each distribution using the 95th percentile and values above that are filtered out in order to avoid skewed results. Figure S5 shows the 95th percentile of the $\Delta x$ as a function of the different a priori sources and for different layers. The lowest values are found for both ERA-d and ERA-6, and about 25 % larger values are found for both NCEP and WACCM.

A quantitative impact of the different a prioris in the retrieval of WV vertical profiles is characterized by means of linear regression and statistical analyses using the layers defined earlier. Since both NOAA FPH and FTIR have uncertainties associated at each altitude we adopted a weighted orthogonal distance regression (ODR) analysis. For a thorough description in weighted ODR applied in atmospheric sciences see Wu and Yu (2018). In order to avoid temporal variability larger than 2 % according to conclusions in section 4.1 a mean WV profile ($\bar{\boldsymbol{x}}_r$) is obtained within a coincidence time interval of 0 - 30 min at BLD and 0

- 60 min for MLO. The NOAA FPH WV mixing rations are used in the abscissa axis and the ODR accounts for uncertainties in both set of measurements. In this case we use the standard deviation of the NOAA FPH and FTIR uncertainty propagated using the individual profiles within the coincident time interval. The final number of vertical profiles used in the comparison are 31 and 30 in BLD and MLO, respectively. Figure 8 shows the slope, intercept, and correlation coefficient (r-value) obtained with the comparison of retrievals using each of the a prioris with the NOAA FPH at different layers in both sites. The error bars in

the estimated parameters are the standard errors. For layers below 10 km the best results are seen with both ERA-I a prioris. In particular, we found that ERA-6 yields the best comparison with a slope close to unity, the lowest intercept, and a correlation coefficient of 0.95 for the layer of 1.5 - 3 km in BLD. For both sites, the second layer, i.e., 3 - 5 and 5.5 - 7.5 km for BLD and MLO respectively, shows lower slopes likely due to gradients between the top planetary boundary layer and free troposphere that are not captured by the retrievals due to coarse vertical resolution and lower sensitivity (e.g., see Figs. 4 and 5).

For each coincidence profile the bias is characterized with the sum of differences between $\bar{x}_r$ and the sonde ($\boldsymbol{x}_s$) profiles divided by the number of points ($N$) in each layer. As described before the number of points in each layer is three. This definition indicates whether the retrievals under - or overestimate the sonde values. The precision is calculated as $2 \times \sigma/\sqrt{N}$, where $\sigma$ is the standard deviation. The bar plot in Fig. 9 shows the median bias and precision in ppm and percentage with respect to the mean values of the NOAA FPH for the different layers and a prioris. The error bars in the bias are estimated

using the $\pm 1 \cdot$standard error of the distribution. The bias shows a high dependency on the a priori. At both sites the first two layers show negative bias for all a prioris. At BLD the smallest bias is found for the 1.5 - 3 km layer with -0.001 $\pm$ 0.105 x$10^3$ ppm (-0.02 $\pm$ 1.86 %) for ERA-6 and the highest bias of -0.27 $\pm$ 0.11 x$10^3$ ppm (-4.82 $\pm$ 1.94 %) for WACCM climatology. The layer between 3 - 5 km shows negative bias between 5.56 % and 11.14 %. Interestingly, NCEP-d yields less biased results in this layer. The layer of 13 - 17 km shows significantly larger values for almost all a prioris (> 15 %). The precision does not

change significantly among different a prioris. The best precision result in percentage is below 5 % found in the lowest layer of 1.5 - 3 km and the highest values of up to 15 % for layers between 5 - 10 km. As expected based in the ODR analysis higher biases are found at MLO. A Negative bias of about 5 % is found for the 3.5 - 5 km layer, and about 10 % for the 5.5 - 7.5 layer and positive 5 % for the 7.5 - 10 km layer. Surprisingly in both sites WACCM yields lower bias for the layers above 13 km. In general, among all layers the lowest bias are found using ERA-6 and ERA-d for both sites. Table 2 presents a summary of the





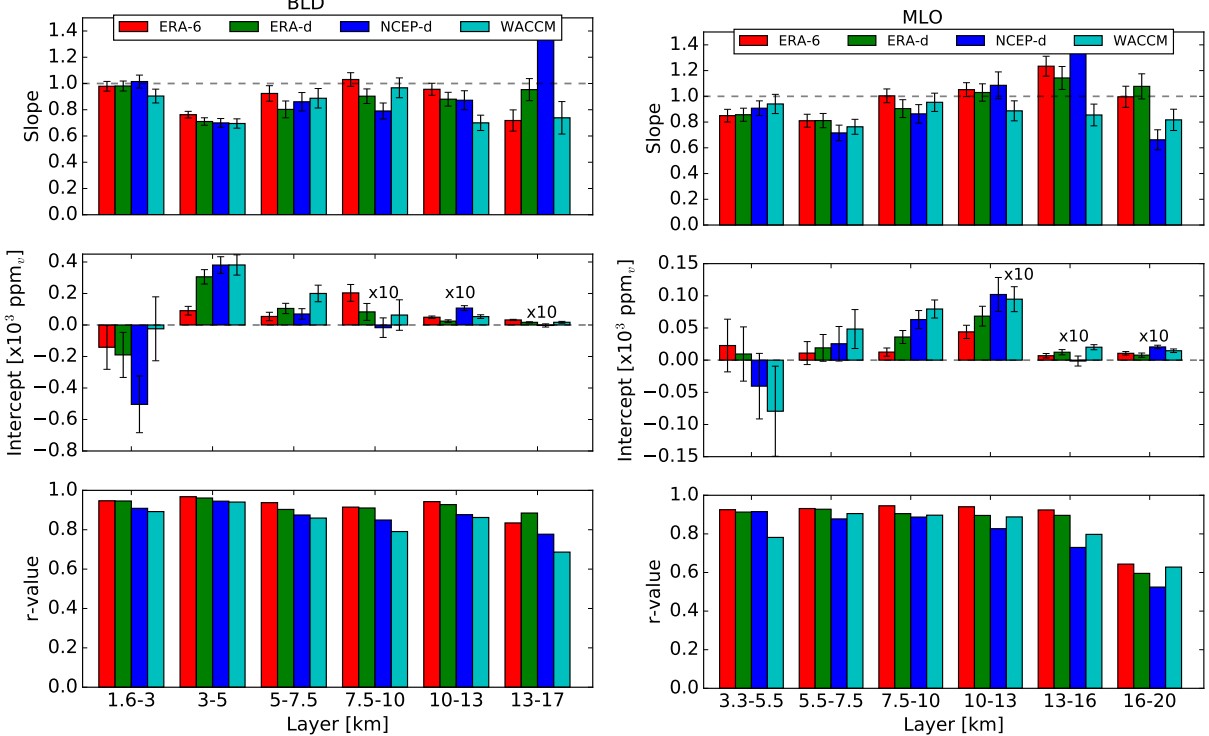

**Figure 8.** Results of the ODR analysis between the NOAA FPH and FTIR using different a priori profiles at different altitude layers. Error bars represent the standard errors of the estimated parameters. Note that for visibility the intercept obtained in the upper three layers has been multiplied by a factor of 10.

ODR and statistical analysis using ERA-6 at BLD where the spatial mismatch is known and the launch of the sonde is in close proximity to the FTIR location.

**Table 2.** Summary of the ODR and statistical analysis using ERA-6 at BLD.

| Layer [km] | slope | Intercept [x$10^3$ ppm] | r-value | Bias [%] | Precision [%] |
|---|---|---|---|---|---|
| 1.6 - 3.0 | $0.98 \pm 0.04$ | $-0.14 \pm 0.14$ | 0.95 | $-0.02 \pm 1.9$ | 3.7 |
| 3.0 - 5.0 | $0.76 \pm 0.03$ | $0.09 \pm 0.03$ | 0.97 | $-11.1 \pm 3.5$ | 7.0 |
| 5.0 - 7.5 | $0.92 \pm 0.06$ | $0.05 \pm 0.03$ | 0.94 | $1.0 \pm 5.3$ | 10.6 |
| 7.5 - 10.0 | $1.03 \pm 0.05$ | $0.02 \pm 0.005$ | 0.91 | $13.0 \pm 5.0$ | 10.0 |
| 10.5 - 13.0 | $0.96 \pm 0.05$ | $0.005 \pm 0.001$ | 0.94 | $13.1 \pm 5.3$ | 10.6 |
| 13.0 -17.0 | $0.72 \pm 0.08$ | $0.003 \pm 0.001$ | 0.83 | $41.6 \pm 4.0$ | 8.1 |





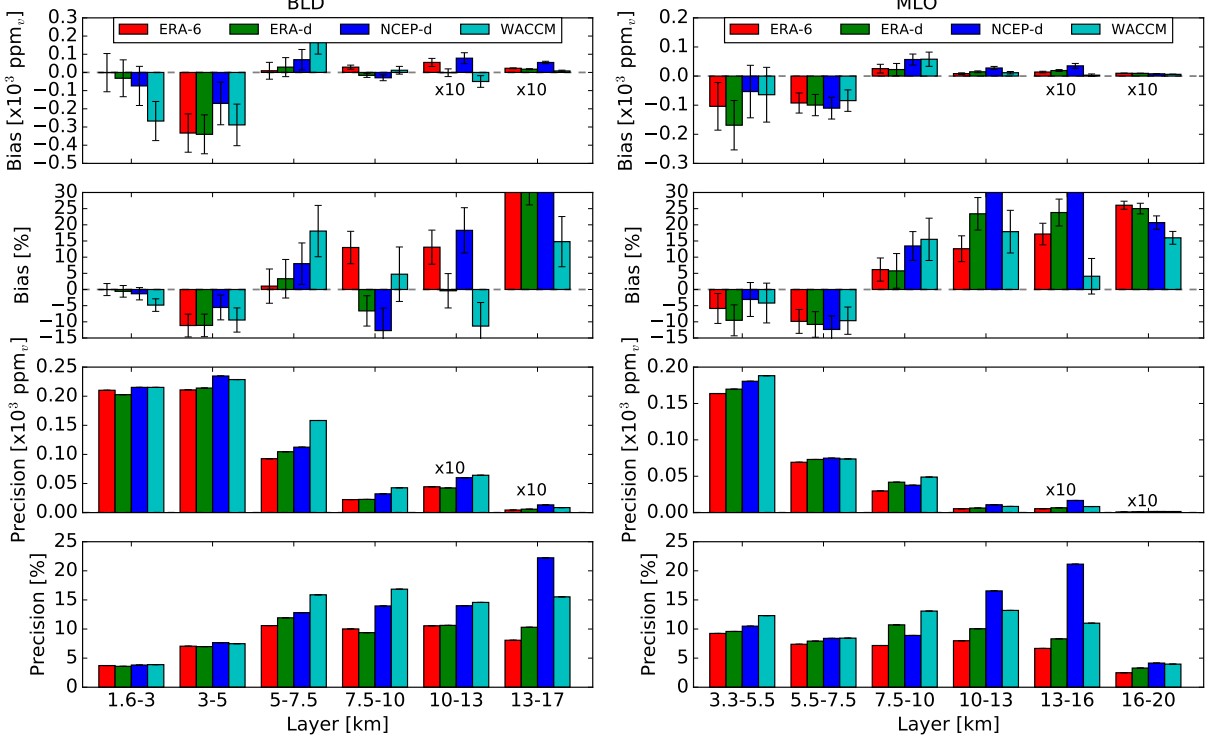

**Figure 9.** Statistical analysis results (bias and precision) of the FTIR WV retrieved at different altitudes and using different a priori profiles for BLD (left) and MLO (right). Bias and precision are given in mixing ratios and percent with respect to the mean values at each layer. The error bars in the bias represent the standard error of the distribution. Note that for visibility the bias and precision in mixing ratio from the two upper layers have been multiplied by a factor of 10.

## 5 Influence of WV on gas profile retrievals

Absorption of WV is normally expected in the analysis of gases using FTIR measurements. Even optimized micro-windows include the WV and/or isotopologues absorption lines in order to minimize interference. In the retrieval process, a vertical profile is fitted normally for a target gas and other species can also be fitted as profile or simply by single scaling of their a priori profile. This a priori or reference profile may play an important role, especially if it changes significantly diurnally and seasonally. In the case of WV sometimes it is retrieved in dedicated micro-windows and then used in the retrieval of other gases. Normally, WV is again fitted, but now with only one scaling parameter (Vigouroux et al., 2012). So far, however, there is little published data on the quantitative impact of the WV a priori profile. In this section we investigate the influence of the different WV sources, i.e., ERA-6, ERA-d, NCEP, WACCM, and retrieved WV profiles, in the retrieval of selected gases. Note that we do not aim to study retrieval strategies of gases or the validation of profile retrievals but rather to show the relative difference with respect to the 'truth' WV profile, in this case the NOAA FPH. Table 3 presents the three target gases (HCN, CO, and $C_2H_6$) and a summary of the retrieval settings. The settings we follow are IRWG/NDACC standard operational retrieval parameters





**Table 3.** Retrieval settings of gases to study the influence of WV. All interfering species are fitted with scaling factor, except $O_3$ in the retrieval of CO and $C_2H_6$ and is fitted as vertical profile.

| Gas | Micro-windows [cm$^{-1}$] | Interfering species |
|---|---|---|
| CO | 2057.7-2058.0; 2069.56-2069.76; 2157.50- 2159.15 | $O_3$, $CO_2$, OCS, $H_2O$, $N_2O$ |
| HCN | 3268.04-3268.40; 3287.10-3287.35; 3299.40-3299.60 | $H_2O$, $C_2H_2$, $CO_2$, $O_3$ |
| $C_2H_6$ | 2976.66-2977.059; 2983.20-2983.50; 2986.45-2986.85 | $O_3$, $H_2O$, $CH_4$, $CH_3Cl$ |

with respect to micro-windows and interfering species. The WACCM climatology is used for a priori profiles of interfering species. Spectroscopic line parameters are adopted from HITRAN 2008 (Rothman et al., 2009, 2013). For the retrieval of HCN we followed a similar approach applied in Paton-Walsh et al. (2010); Vigouroux et al. (2012); Viatte et al. (2014). The settings applied in the CO retrieval are part of an ongoing project in the IRWG of NDACC (B. Langerock, personal communication,

2017), and for $C_2H_6$ we applied an improved version applied in Franco et al. (2015) (E. Mahieu, personal communication, 2017). Pressure and temperature profiles are from NCEP at NDACC. For the retrieval of WV we use ERA-d to imitate our typical retrieval strategy.

The retrieval of HCN, CO, and $C_2H_6$ was performed only during dates with NOAA FPH sonde measurements. Since the FPH profiles are taken as the ground 'truth' we have limited spectra taken only within 1 h of the sonde launch based on findings

presented earlier. In all cases, the standard settings remain the same and only the WV profile reference is changed. An example of the effect of WV profile in the retrieval of the different gases is shown in Figure 10. The different WV profiles used on this day (July 22 2014) are shown on top. The retrieved WV (black) is the closest in shape and magnitude to the NOAA FPH profile (purple). All the other WV profiles show significant differences with respect to the FPH. The gas profile retrievals are shown on the left panels using similar color scheme as in the WV profile panel. The relative difference at every retrieval level, defined

as $(x_i - x_{fph})/x_{fph} \times 100$, is shown on the right panels. The lowest relative difference in all grid points and for all gases is always when using the retrieved WV profile (black). All other WV sources present significant differences. For example, for HCN differences of up to -20 % are found at 6-10 km if using ERA-I. CO and $C_2H_6$ also show important differences but always below 10 %. This example suggests that the current retrieval strategy of WV is suitable for obtaining WV vertical profiles and will improve the retrieval of other gases.





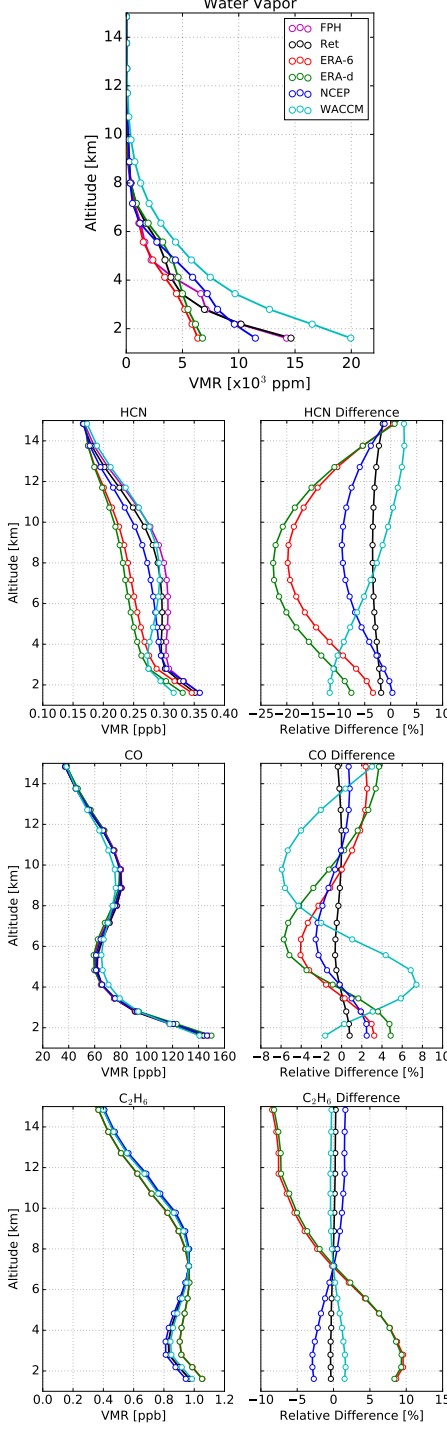

**Figure 10.** Example on July 22 2014 of retrieval profiles of the HCN, CO, and $C_2H_6$ using the different WV a priori sources shown on top. The retrieval profiles are left and right panels represent the relative different in percent with respect to the retrieval, which uses NOAA FPH WV.




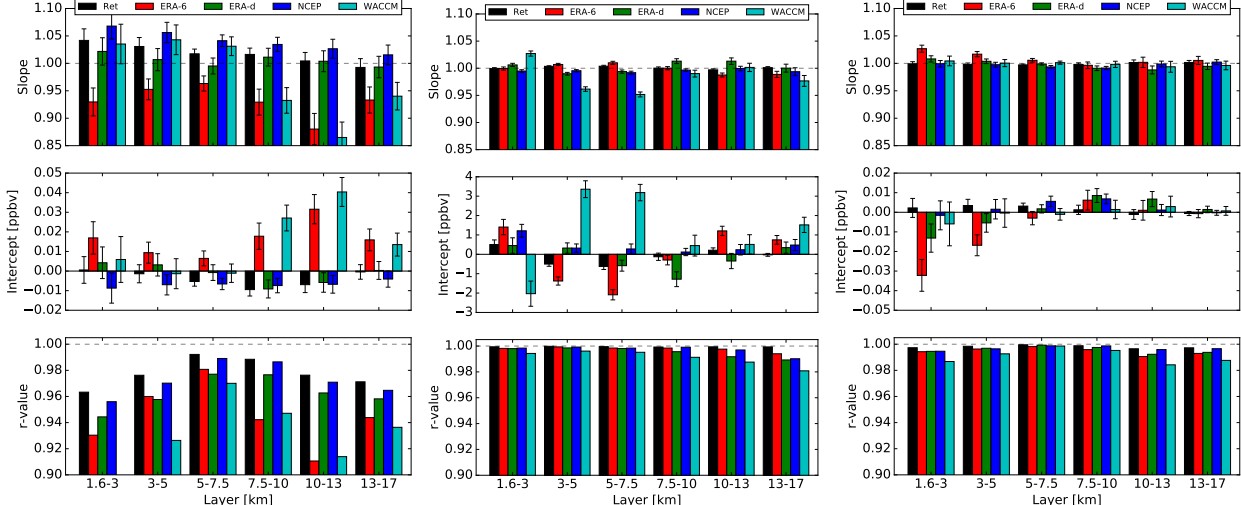

**Figure 11.** Results of the ODR analysis where the mixing ratios using different WV sources at different layers are compared with the 'truth' retrieved values using the NOAA FPH WV for HCN (left), CO (middle), and $C_2H_6$ (right). Error bars represent the standard errors of the estimated parameters.

In order to determine the general impact of the different WV sources for all spectra recorded within 1 h of sonde launch for 6 years we have performed an ODR and statistical analysis similar to the one presented in section 4.3. In this case, the retrieval using NOAA FPH WV is used as the reference. Figure 11 shows the main results of the ODR analysis for the three gases using the different WV sources, and at different layers. The best correlations (r-value) and the lowest intercepts are found using the

pre-retrieved WV profiles for all three gases, in agreement with the example given in Fig. 10. The slope values are close to unity and within the uncertainty values for CO (middle) and $C_2H_6$ (right) using the pre-retrieved WV. However, HCN on the left shows the most notable difference with respect to unity. The intercept is normally negligible for pre-retrieved WV for all gases. The bias and precision results are shown in Fig. 12. A bias larger than 6 and 1 % are found for HCN and CO respectively using WACCM WV in the layer close to the surface. $C_2H_6$ does not show a significant bias among different layers and WV

sources.

## 6   Conclusions

The aim of the present research was to determine the limitations to retrieve real WV structural variability from the boundary layer to the upper troposphere - lower stratosphere using a standard FTIR inversion, i.e., a current retrieval strategy is not further improved to correlate well with reference vertical profiles. Highly precise and accurate vertical profiles of WV from

NOAA balloon FPH in-situ sondes are used for the first time as reference to evaluate FTIR WV profiles in BLD and MLO allowing the characterization of the retrievals in mid-latitudes boundary layer and sub-tropical free troposphere.





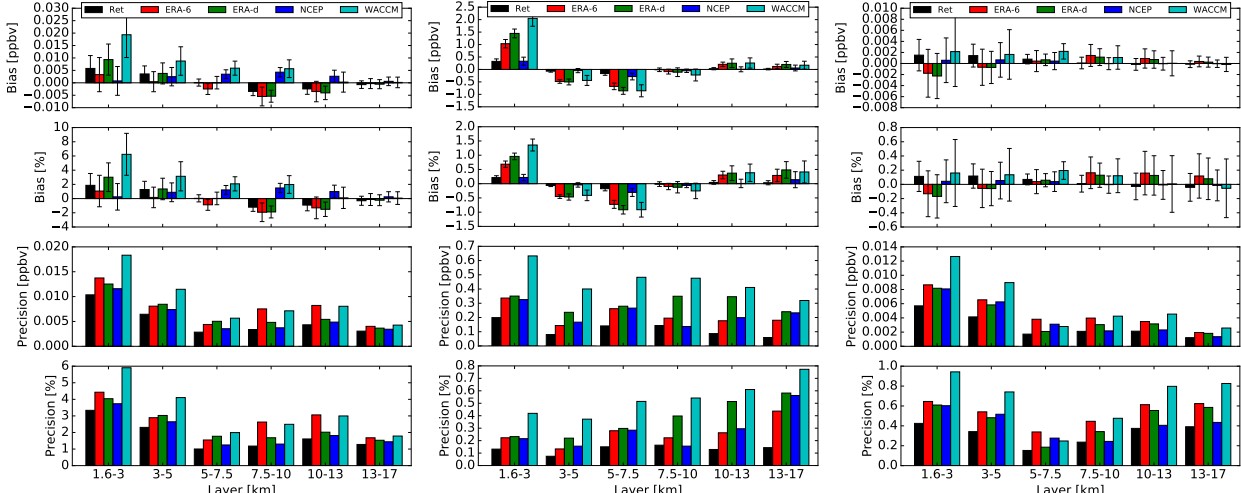

**Figure 12.** Statistical analysis results (bias and precision) for HCN (left), CO (middle), and $C_2H_6$ (right) using different WV profiles at different altitudes. The error bars in the bias represent the standard error of the distribution.

The spatial-temporal variability of WV is inferred prior to a quantitative comparison. By using daily continuous FTIR measurements we derive a temporal variability for different altitudes and find that at BLD the different layers are highly correlated and show comparable variability. In contrast, at MLO the variability among layers is quite different indicating vigorous inhomogeneity due to local convection or long-range transport. The ideal coincidence time between sonde launch and FTIR measurements are 0 - 30 min and 0 - 60 min in BLD and MLO, respectively to avoid variability larger than 2 % for all altitudes. The horizontal position with maximal sensitivity of WV distribution is derived for each FTIR measurement. Then, based on the sonde location at each altitude the horizontal spatial mismatch is characterized. The insights gained from this evaluation is that the boundary layer (about 1.5 to 3 km in Boulder) is the only layer where the air mass probed by the FTIR and NOAA FPH in-situ is likely unchanged since the horizontal difference remains below 10 km. We show that above 5 km the spatial mismatch increased significantly up to 60 km horizontal distance at about 10 km altitude. This feature does not depend on the coincidence time between measurements but rather on the local to synoptic meteorological scales. More broadly, even co-located FTIR and sonde launch measurements would have significant horizontal mismatch at different altitudes. Further work needs to be done to establish the best methodology to validate FTIR profile retrievals while avoiding difference in geometry of measurements.

This work offers a new assessment of the accuracy and precision of FTIR retrievals at different altitudes. The analysis consists of the comparison of WV using several atmospheric layers using a ODR and statistical analysis, i.e., estimation of accuracy and precision. Furthermore, we study the effect of different WV a priori commonly used among NDACC stations (ERA-I, NCEP, and WACCM sources). The following overall conclusions can be drawn from the un-smoothed comparison of WV using several layers: (1) using 6 hourly and daily ERA-I a priori shows the best correlation and comparison in both sites; (2) the lowest bias and precision are found in the closest layer to the instrument (1.5 - 3km at BLD and 3 - 5 km at MLO).



At BLD, we report a negligible negative bias of -0.001 $\pm$ 0.105 $\times 10^3$ ppm (-0.02 $\pm$ 1.9 %) and precision of 0.21 $\times 10^3$ ppm (3.7 %) for the 1.5 - 3 km layer while at MLO the bias is -0.10 $\pm$ 0.08 $\times 10^3$ ppm (-5.8 $\pm$ 4.6 %) and precision of 0.16 $\times 10^3$ ppm (9.2 %) for the 3 - 5.5 km layer, which are larger likely due to the significant spatial mismatch difference between the location of measurements; (3) high vertical variability probed by the sonde in the second layer is not fully captured by the

retrievals, although is considerably better than a priori profiles; (4) and one significant findings to emerge is that the retrievals show encouraging results in the 10.5 - 13.5 km at BLD and 13 - 16 km at MLO (roughly the UTLS layer) with 13.1 $\pm$ 5.3 % (BLD) bias and a precision of 10.6 % (BLD) but the bias increases to about 40 % above this layer. Table 2 was constructed to show a representative analysis when the spatial mismatch is known and when the location of the FTIR and the launch of the sonde are near each other. According to these results we infer that the interpretation of the averaging kernels and degrees

of freedom are quite conservative and WV retrievals contain more information than expected. The findings of this study show that FTIR profiles can be used to evaluate long-term records of WV at several partial columns in the troposphere.

    The second goal of this study was to investigate the influence of WV in the retrieval of other tropospheric gas profiles with DOF larger than two. Here we present results for three important gases, i.e., HCN, CO, and $C_2H_6$ using the WV NOAA FPH profile as ground 'truth' as reference and comparing to other WV sources, including the retrieved WV, ERA-I, NCEP, and

WACCM. In general, our results recommend retrieving WV profiles first then using them as input to the retrievals of other gases in order to reduce bias due to imperfect WV vertical profile. As an example (Fig. 10) we show relative differences of up to 25 % at 8 km, 8 % at 4 km, and 10 % at 3 km for HCN, CO, and $C_2H_6$ if WV is not retrieved beforehand and used posterior as the input profile. Overall, a statistical comparison of all profiles in the 1.5 - 3.0 km show significant impact on HCN (about 6 % bias), middle impact in CO (about 1.2 % bias), and low impact on $C_2H_6$ (< 0.5 % bias). This sensitivity study is the first

comprehensive quantitative investigation in this topic and provides a basis for future error budget assessment. In principle we hypothesize that the effect of WV profiles might be bigger in humid sites within the boundary layer but further research should be carried out to establish its quantitative importance.

## 7   Data availability

The NCAR FTIR water vapor retrievals can be obtained from the authors upon request. Vertical Profile of Water Vapor from

Balloon flight NOAA can be accessed through the website: https://www.esrl.noaa.gov/gmd/dv/data/index.php?parameter_name=Water%2BVapor.

*Disclaimer.* The National Center for Atmospheric Research is sponsored by the National Science Foundation. Any opinions, findings, and conclusions or recommendations expressed in this publication are those of the author(s) and do not necessarily reflect the views of the National Science Foundation.

*Acknowledgements.* This study has been supported under contract by the National Aeronautics and Space Administration (NASA). We are grateful to the NOAA staff at MLO for technical support and maintenance of the NCAR FTIR. Especially, we wish to thank Paul Fukumura.



We would like to thank David Nardini and Darryl Kuniyuki for diligently preparing and launching the NOAA FPH instruments monthly from Hilo, Hawaii. We thank Helen Worden for her valuable suggestions during the NCAR internal review.





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
