# Peer review of "Tropospheric water vapor profiles obtained with FTIR: comparison with balloon-borne frost point hygrometers and influence on trace gas retrievals"

_Atmospheric Measurement Techniques, 2018_

## Referee Comment (RC1) · Anonymous Referee #2 · 13 Nov 2018

**1   General Comments**

The paper deals with two main topics:

a)  first the authors assess the limitations in retrieving the real Water Vapor (WV) vertical variability from the boundary layer to the upper troposphere - lower stratosphere, with a standard inversion of FT solar absorption measurements in the middle-infrared.  The study includes the validation of WV profiles retrieved from

ground based FTS measurements operated from Boulder (Colorado) and Mauna-Loa (Hawaii), via intercomparison with WV profiles measured by state-of-the-art Frost Point Hygrometers (FPH) operated from balloons.

b) Secondly, a sensitivity study is presented, showing the error on retrieved HCN, CO, and $C_2H_6$ VMRs due to assuming a less than perfect WV vertical profile.

The subject of the paper is clearly within the scope of AMT. The methods used are scientifically sound, the presentation is sufficiently concise, however it could be improved by rephrasing a few sentences as outlined in the specific comments reported below. The paper does not introduce novel concepts or ideas, however the results of the study will be useful for other scientists using the data presented or data deriving from similar measurements. For this reason I recommend this paper for publication in AMT, after some revisions as outlined below.

My main comment or criticism is about the strategy the authors adopt to deal with the Averaging Kernels (AKs). I agree that the AKs may not be a sufficiently accurate tool to evaluate the smoothing error of the retrieved WV profiles. This is due both to the fact that AKs are only a "linear" approximation of the vertical response function of the measuring system (instrument plus retrieval algorithm), and to the fact that it is generally hard to setup a covariance matrix which represents properly the variability of WV from ground to the Upper-Troposphere / Lower Stratosphere (UTLS). To show the limitations (in your test case) of the smoothing error as derived from AKs and the Rodgers (2000) approach, rather than moving the AKs analysis to the supplemental material, I would have compared, in the main paper, the actual smoothing error (obtained via intercomparisons with FPH) with the estimate of the same error obtained from AKs.

The second general comment I have is about the sensitivity analysis presented in Sect. 5. To my opinion it would be worth to better explain why, after the analysis presented in the first part of the paper, then you start to study a quite different subject, such as the mapping of WV errors on subsequent VMR retrieval of other gases. More-

over, since your measurements cover the middle-infrared, I also expect a sensitivity of the retrieved VMRs to the temperature error. This is already shown in Fig. 3 for WV. What about the error on HCN, CO, and $C_2H_6$ VMRs due to the temperature error ? Do you suggest to retrieve also the temperature profile from the same measurements or you are satisfied with the temperature profiles taken from NCEP at NDACC ?

**2  Specific Comments**

P4 L27,28: Constraining is important to select the solution which, among the possible solutions of the ill-posed inversion, is the most likely on the basis of our prior knowledge.

P4 Eq.1: Here it is not clear if your retrieval performs only a single or several Gauss-Newton iterations, because you don't have an iteration index in the Equation. Please, also define clearly the meaning of $\mathbf{K}$. Do you compute it at each iteration ? I guess $\mathbf{K}$ is the Jacobian of the forward model with respect to the retrieval parameters, therefore it should be re-computed at each iteration and should show an iteration index.

P5 L2: It would be interesting here to know which is the spectral range covered by the spectrometers used, and which is the rationale behind the selection of the listed micro-windows (e.g. minimum retrieval error ?).

P5 L8,9: I got an idea of what the authors would like to say, however I suggest to re-phrase more clearly this sentence.

P5 L10: Which are the "relaxed covariance matrices" that induce oscillations ? Please clarify.

P5 L14: If apodization is not used, how broad is the ILS used in the forward model to emulate the instrument effect ?

P5 L25,26: Here it is not clear how the 0.5% rms error "on the fit" maps onto the retrieved WV. Does this rms error refer to the "residuals of the fit" or directly to the retrieved WV ?

P6 Fig.1: It would be better to show $CH_4$ and $N_2O$ absorption contributions with different line colours.

P6 L14-16: This bias is with respect to the a-priori state vector which, in turn, will probably have some bias with respect to the real profile. Note that if the bias of the retrieval was known both in sign and amplitude, then it would be possible to correct for it...

P7 L1-5: Here I would state clearly which is your retrieval vector. Do you retrieve a WV profile using the discretization mentioned in the colour scale of Fig.2a ? Do you include further fitting parameters ? (Such as atmospheric continuum, for example).

P7 L7-9: Smoothing the high-vertical-resolution profiles via the averaging kernels of the coarse-vertical-resolution experiment is not mandatory, especially if you attribute a "smoothing error" to the profile differences or if you want to try characterizing the smoothing error itself. Therefore I would simply state your choice here, without trying to find a justification, which is also rather fuzzy to my view.

P7 L18: Off-diagonal elements of $\mathbf{S}_e$ may play a very important role if the spectrum is oversampled (wrt interferogram) and/or if apodization is used. Please state explicitly that, apparently, this is not your case.

P8 L8: Please define also the symbol $\mathbf{K}_b$.

P8 L17-ff: In Fig.3a the error due to interfering species is also shown. Which are the considered interfering species that are not simultaneously retrieved with WV ?

P9 L1: This sentence is not clear and may be questionable. Does this mean that your AKs are not a good estimate of the vertical response function of your system (instrument plus inversion scheme) ? Why ?

P9 L7: Please note that re-gridding via interpolation is, on its own, an arbitrary smoothing. So, I do not fully understand why you do not want to use AKs to smooth and re-sample high-resolution profiles prior to intercomparison (as it seems you already did in the plots presented in the supplemental material).

P9 L12-14: As shown in Fig. 2c, the FTS retrievals have less than 3 DOFs therefore, why using so many layers for the intercomparison ? The risk is to find biases of different sign in adjacent layers.

P10 Sect. 4.1: The underlying idea is good, however, please note that the variability evaluated here could underestimate the real WV variability, due to the constraint of the retrievals towards the a-priori state vactor. I would have estimated from measurements the variability of the spectrum vs time and would have derived the corresponding "time-mismatch" error covariance matrices relating to the individual WV profiles, using Eq. 3. I suggest to include a comment on this regard (or change approach...).

P11 Sect. 4.2: Here I did not understand if, from this analysis, you also derive an estimate of the error component to be attributed to the difference between FTS and sonde WV profiles, due to the spatial mismatch of the measurements. I agree that it is hard to derive such an error estimate however, lacking this estimate, I do not see very much the usefulness of this section. Please explain.

P13 L2,3: The second effect of a-priori WV on the solution is not clear. Did-you mean that the a-priori WV influences the solution also because it is used as initial guess for the Gauss-Newton iterations ? (This latter effect should be negligible if the retrieval converges properly).

P16, Sect. 5: the link between this Section and the work presented in the previous Sections of the paper is non very clear. Maybe you could state at the beginning of this Section how the work you are going to present is linked with the analysis presented earlier.

P16 L1,2: What is the meaning of "expecting WV" in this context ? Usually an optimized microwindow selection scheme tries to avoid spectral interferences from WV and related isotopologues. From the second sentence, however, I understand the opposite.

P17 Table 3: Which is the rationale for the adopted sorting of interfering species in the Table ? Perhaps their relative importance ? In this case one should assess the retrieval errors due to the interference of ozone. In this spectral region I also expect temperature knowledge to be of importance (see general comments above). Moreover, I understand that the micro-windows used were selected in already publishes papers, however you could at least mention here the rationale with which they were selected / optimized (e.g. with the aim of minimizing the total retrieval error of the gas to be retrieved).

**3 Technical Corrections**

P2 L26: Remove double comma between "Zugspitze" and "Germany".

P4 L3: Description... described elsewhere (rewording needed).

P7 L3: but as explained before, one of the goals...

P11 Caption of Fig. 5: Same as Fig. 4 but for MLO.

P12 L9: As mentioned above, the initial spatial difference...

P13 L17: I do not see the usefulness of putting some figures in the supplemental material when these are recalled and described in the text of the main paper. I would put all the figures in the main paper file (of course only if this operation does not cost too much!)

P16 Fig. 9: in the vertical axis labels "ppmv" has a small "v", is this an intentional choice ?

---

## Referee Comment (RC2) · Schneider (Referee) · 13 Dec 2018

Dear Editor,
Dear Authors,

the manuscript addresses the following topics:

(1) it describes the retrieval of water vapour profile from ground-based FTIR measurements, (2) it compares the retrieval data with frost point hygrometer sonde data, (3) it assess the impact of different WV a priori data on the retrieval of water vapour, and (4) it assess the impact of different WV a priori data on the retrieval of other trace gases where water vapour is an important interfering species.

My general comments:

I find (4) is a nice and valuable demonstration of the importance for using actual WV profile data in order to avoid large uncertainties in the retrievals of other trace gases. The reason is that WV is very variable and not well capturing the variability results in large retrieval errors of the other species. However, this part of the paper could be further improved by inserting references on previous work where the interference error of WV has been calculated.

I think (1)-(3) need revisions. A constrained remote sensing data product (here $x_r$) means that a priori data (here $x_a$) has been updated with a measurement. The product ($x_r$) strongly depends on the a priori data ($x_a$). In particularly if $x_a$ is variable on small scales (like for WV) the variability in $x_r$ will, to a large extent, reflect the variability of the prescribed $x_a$. Instead of assessing the quality of $x_r$ the authors should assess how the remote sensing measurement can improve the assumed a priori state of the atmosphere, i.e. the authors should validate $x_r - x_a$ by comparing it to $A(x_s - x_a)$, where $x_s$ is the FPH reference.

Furthermore, when using an a priori data that already captures most of the variability, the solution state should be more constrained (the $S_a$ matrix should have much smaller entries) than when using an $x_a$ that captures only few variability. However, judging from Sect. 3 it seems that the authors use a single $S_a$ for constraining the different retrievals.

Specific comments:

I have inserted my ideas/suggestions in the attached pdf version of the manuscript.

Best regards.

[revised manuscript text omitted]

**Summary of Comments on amt-2018-283_MS.pdf**

**Page: 2**

| | | | |
|---|---|---|---|
| Number: 1 | Author: pa5682 | Subject: Cross-Out | Date: 12/3/2018 8:01:36 PM |

| | | | |
|---|---|---|---|
| Number: 2 | Author: pa5682 | Subject: Inserted Text | Date: 12/3/2018 7:55:01 PM |

uses

| | | | |
|---|---|---|---|
| Number: 3 | Author: pa5682 | Subject: Inserted Text | Date: 12/3/2018 10:25:15 PM |

/FTIR spectra measured at 12 different sites

| | | | |
|---|---|---|---|
| Number: 4 | Author: pa5682 | Subject: Inserted Text | Date: 12/3/2018 10:36:31 PM |

for generating a long-term data set of global representativeness of tropospheric water vapour profiles with a DOFS of almost 2.8 and of about 1.6 for the ratio between the most abundant isotopologue $H_2^{16}O$ and the heavy isotopologue $HD^{16}O$.

| | | | |
|---|---|---|---|
| Number: 5 | Author: pa5682 | Subject: Inserted Text | Date: 12/3/2018 8:03:40 PM |

Comparisons of FTIR and operational radiosondes have been
used to validate optimized WV profile retrieval strategies (e.g. Schneider et al. 2006; Schneider and Hase, 2009; Schneider et al., 2016).

| | | | |
|---|---|---|---|
| Number: 6 | Author: pa5682 | Subject: Inserted Text | Date: 12/3/2018 10:37:43 PM |

made at two different sites, that have so far not been considered within MUSICA. We use spectral microwindows that are not identical to those of MUSICA (Barthlott et al., 2017, Fig. 1 therein) 
[revised manuscript text omitted]

For constraining the autors use the same $S\_a^{-1}$ for the different a prioris from Section 4.3?

If yes, the solution state will be much looser constraint for the daily varying a priori than for the monthly varying a priori, i.e the two retrievals are dificult to be compared.

[Figure]

[Figure]

**Figure 1.** Mean retrieval fit between 2010 - 2016 for the spectral intervals of WV. The observed and fitted lines are blue and green respectively. The absorption contribution for the different species are also shown in each micro-window. The bottom black lines represent the mean residual and the gray shadow are the standard deviation. Note that for visibility the residuals have been multiplied by 10.

data and the bias of the fits are insignificant indicating that the daily mean temperature profile is adequate for the retrievals. [1]With the exception of WV (see section 4.3), VMR input mean profiles of all other gases are taken from the 40 year run of WACCM.

**3.1 Characterization and error budget**

The mean retrieval fit of the four micro-windows between 2010-2016 at BLD is shown in Fig. 1. The small systematic residual structures (black lines) are likely caused by spectroscopic parameter error but in general the magnitude of residuals is low and within noise level ($< 0.1 \%$).

The information content of the retrieved WV vertical profile is characterized by means of the averaging kernel matrix, $\mathbf{A}$:

$$\mathbf{A} = \left( \mathbf{K}^T \mathbf{S}_e^{-1} \mathbf{K} + \mathbf{S}_a^{-1} \right)^{-1} \mathbf{K}^T \mathbf{S}_e^{-1} \mathbf{K} \qquad (2)$$

The rows of the mean $\mathbf{A}$ known as averaging kernels (AK) at BLD obtained between 2010 - 2016 color coded by altitude below 20 km are shown in Fig. 2a. The maximum values are located at the surface, then they decrease and remain steady to about 8 km and eventually decrease to zero above 12 km. This indicates that most of the information content obtained from WV pressure dependence of the absorption features is limited to the troposphere. The mean total column averaging kernel (TAK) is shown in Fig. 2b. Tipically, a unity TAK indicates that the retrieval is not biased, while values of the TAK lower than unity indicate underestimation and larger values than unity indicate overestimation of the real WV. Hence, below 3 km the retrieval may underestimate, between 3-8 km overestimate, and between 8-12 km underestimate the real WV magnitude. The mean number of degrees of freedom (DOF), given by the trace of the $\mathbf{A}$, are 2.4 and indicate the total number of independent pieces

It is important to describe here the remote sensing measurement as an update of the a priori information, i.e. the actual measurement is $x_r - x_a$ not $x_r$! The authors should explain how the a priori affect the retrieval results: $x_r = A(x - x_a) + x_a + Dx$, where $Dx$ are retrieval errors. Because A is not an identity matrix $x_r$ will always significantly depend on $x_a$.

The authors use variable a priori data (see Sect. 4.3) and the variability seen in the retrieved vertical profile reflects to large extent the variability prescribed by the a priori data. Unfortunately this is not correctly considered in Sect 4 of the paper. Because the authors work with a variable a priori they should evaluate the signals in $x_r - x_a = A(x - x_a)$, because this is the mesured signal not $x_r = A(x - x_a) + x_a$! Furthermore, the authors assume that if $x_s$ can be used as a reference for the retrieved profile $x_r$; howrever, actually $x_s$ is highly resolved and absolutely calibrated reference, i.e. it is a reference for the real atmospheric profile x. So correcty the authors should compare $x_r - x_a$ with $A(x_s - x_a)$ in order to validate the remote sensing measurement.

[revised manuscript text omitted]

Number: 1    Author: pa5682    Subject: Highlight   Date: 12/4/2018 2:36:40 PM

$x\_r-x\_s=(I-A)*(x\_a-x\_s)+Dx$, i.e. it depends on the retrieval errors Dx and on $(x\_a-x\_s)$. Because a daily or even 6 hourly varying $x\_a$ better captures the actual variability of atmospheric WV, using $x\_a$ from ERA-d and ERA-6 (instead of monthly climatologies) better captures the variability as given in $x\_s$, i.e. $x\_r-x\_s$ shows a particular small scatter. This is no surprise.

[Figure]

[Figure]

**Figure 8.** Results of the ODR analysis between the NOAA FPH and FTIR using different a priori profiles at different altitude layers. Error bars represent the standard errors of the estimated parameters. Note that for visibility the intercept obtained in the upper three layers has been multiplied by a factor of 10.

ODR and statistical analysis using ERA-6 at BLD where the spatial mismatch is known and the launch of the sonde is in close proximity to the FTIR location.

**Table 2.** Summary of the ODR and statistical analysis using ERA-6 at BLD.

| Layer [km] | slope | Intercept [x$10^3$ ppm] | r-value | Bias [%] | Precision [%] |
|---|---|---|---|---|---|
| 1.6 - 3.0 | $0.98 \pm 0.04$ | $-0.14 \pm 0.14$ | 0.95 | $-0.02 \pm 1.9$ | 3.7 |
| 3.0 - 5.0 | $0.76 \pm 0.03$ | $0.09 \pm 0.03$ | 0.97 | $-11.1 \pm 3.5$ | 7.0 |
| 5.0 - 7.5 | $0.92 \pm 0.06$ | $0.05 \pm 0.03$ | 0.94 | $1.0 \pm 5.3$ | 10.6 |
| 7.5 - 10.0 | $1.03 \pm 0.05$ | $0.02 \pm 0.005$ | 0.91 | $13.0 \pm 5.0$ | 10.0 |
| 10.5 - 13.0 | $0.96 \pm 0.05$ | $0.005 \pm 0.001$ | 0.94 | $13.1 \pm 5.3$ | 10.6 |
| 13.0 -17.0 | $0.72 \pm 0.08$ | $0.003 \pm 0.001$ | 0.83 | $41.6 \pm 4.0$ | 8.1 |

**Page: 15**

Here retrieval data generated by using varying a priori data are compared to the sonde measurements. This does not allow robust conclusions on the quality of the FTIR measurements. Actually there will be a very good agreement already by comparing the varying a priori with the sonde measurement (i.e. without any information from the FTIR measurement).

What you need to compare and evaluate is the difference with respect to the apriori! So you have to calculate x_r-x_a and correlate it to A*(x_s-x_a).

[Figure]

[Figure]

**Figure 9.** Statistical analysis results (bias and precision) of the FTIR WV retrieved at different altitudes and using different a priori profiles for BLD (left) and MLO (right). Bias and precision are given in mixing ratios and percent with respect to the mean values at each layer. The error bars in the bias represent the standard error of the distribution. Note that for visibility the bias and precision in mixing ratio from the two upper layers have been multiplied by a factor of 10.

**5 Influence of WV on gas profile retrievals**

Absorption of WV is normally expected in the analysis of gases using FTIR measurements. Even optimized micro-windows include the WV and/or isotopologues absorption lines in order to minimize interference. In the retrieval process, a vertical profile is fitted normally for a target gas and other species can also be fitted as profile or simply by single scaling of their a

5    priori profile. This a priori or reference profile may play an important role, especially if it changes significantly diurnally and seasonally. In the case of WV sometimes it is retrieved in dedicated micro-windows and then used in the retrieval of other gases. Normally, WV is again fitted, but now with only one scaling parameter (Vigouroux et al., 2012). So far, however, there is little published data on the quantitative impact of the
[Figure]
 WV a priori profile. In this section we investigate the influence of the different WV sources, i.e., ERA-6, ERA-d, NCEP, WACCM, and retrieved WV profiles, in the retrieval of selected gases. Note that we

10    do not aim to study retrieval strategies of gases or the validation of profile retrievals but rather to show the relative difference with respect to the 'truth' WV profile, in this case the NOAA FPH. Table 3 presents the three target gases (HCN, CO, and $C_2H_6$) and a summary of the retrieval settings. The settings we follow are IRWG/NDACC standard operational retrieval parameters

There is some work:

The impact of interferences from WV on the retrieval of CO has been estimated by Sussmann and Borsdorff, 2007 (doi:10.5194/acp-7-3537-2007), on the retrieval of O3 by García et al. 2014 (doi:10.5194/amt-7-3071-2014) and on the retrieval of CH4 by Sepúlveda et al., 2014 (doi:10.5194/amt-7-2337-2014).

[Figure]

**Table 3.** Retrieval settings of gases to study the influence of WV. All interfering species are fitted with scaling factor, except $O_3$ in the retrieval of CO and $C_2H_6$ and is fitted as vertical profile.

| Gas | Micro-windows [cm$^{-1}$] | Interfering species |
|---|---|---|
| CO | 2057.7-2058.0; 2069.56-2069.76; 2157.50- 2159.15 | $O_3$, $CO_2$, OCS, $H_2O$, $N_2O$ |
| HCN | 3268.04-3268.40; 3287.10-3287.35; 3299.40-3299.60 | $H_2O$, $C_2H_2$, $CO_2$, $O_3$ |
| $C_2H_6$ | 2976.66-2977.059; 2983.20-2983.50; 2986.45-2986.85 | $O_3$, $H_2O$, $CH_4$, $CH_3Cl$ |

with respect to micro-windows and interfering species. The WACCM climatology is used for a priori profiles of interfering species. Spectroscopic line parameters are adopted from HITRAN 2008 (Rothman et al., 2009, 2013). For the retrieval of HCN we followed a similar approach applied in Paton-Walsh et al. (2010); Vigouroux et al. (2012); Viatte et al. (2014). The settings applied in the CO retrieval are part of an ongoing project in the IRWG of NDACC (B. Langerock, personal communication,
5 2017), and for $C_2H_6$ we applied an improved version applied in Franco et al. (2015) (E. Mahieu, personal communication, 2017). Pressure and temperature profiles are from NCEP at NDACC. For the retrieval of WV we use ERA-d to imitate our typical retrieval strategy.

The retrieval of HCN, CO, and $C_2H_6$ was performed only during dates with NOAA FPH sonde measurements. Since the FPH profiles are taken as the ground 'truth' we have limited spectra taken only within 1 h of the sonde launch based on findings
10 presented earlier. In all cases, the standard settings remain the same and only the WV profile reference is changed. An example of the effect of WV profile in the retrieval of the different gases is shown in Figure 10. The different WV profiles used on this day (July 22 2014) are shown on top. The retrieved WV (black) is the closest in shape and magnitude to the NOAA FPH profile (purple). All the other WV profiles show significant differences with respect to the FPH. The gas profile retrievals are shown on the left panels using similar color scheme as in the WV profile panel. The relative difference at every retrieval level, defined
15 as $(x_i - x_{fph})/x_{fph} \times 100$, is shown on the right panels. The lowest relative difference in all grid points and for all gases is always when using the retrieved WV profile (black). All other WV sources present significant differences. For example, for HCN differences of up to -20 % are found at 6-10 km if using ERA-I. CO and $C_2H_6$ also show important differences but always below 10 %. This example suggests that the current retrieval strategy of WV is suitable for obtaining WV vertical profiles and will improve the retrieval of other gases.

Number: 1     Author: pa5682     Subject: Highlight   Date: 12/7/2018 6:45:22 PM
The strategy is sufficient to avoid WV interferences in the retrievals of other trace gases and the obtained WV profiles are of a reasonable quality. However, using retrievals with more WV lines, and retrievals on log scale (tropospheric water vapour is log-normally distributed) should theoretically provide better results. Actually the MUSICA WV data show higher DOF and agreement with radiosonde also within 10-20%.

[revised manuscript text omitted]

---

## Author Comment (AC1) · 10 Jan 2019

Response to Anonymous Referee #2; Dec 2018

Black: Referee's comments
Green: Author's reply

We would like to thank Reviewer #2 for taking the time to review our manuscript. We believe the revised manuscript has improved thanks to thorough and thoughtful comments provided.

**1 General Comments**

The paper deals with two main topics:

a) first the authors assess the limitations in retrieving the real Water Vapor (WV) vertical variability from the boundary layer to the upper troposphere - lower stratosphere, with a standard inversion of FT solar absorption measurements in the middle-infrared. The study includes the validation of WV profiles retrieved from round based FTS measurements operated from Boulder (Colorado) and Mauna- Loa (Hawaii), via intercomparison with WV profiles measured by state-of-the-art Frost Point Hygrometers (FPH) operated from balloons.

b) Secondly, a sensitivity study is presented, showing the error on retrieved HCN, CO, and C2H6 VMRs due to assuming a less than perfect WV vertical profile.

The subject of the paper is clearly within the scope of AMT. The methods used are scientifically sound, the presentation is sufficiently concise, however it could be improved by rephrasing a few sentences as outlined in the specific comments reported below. The paper does not introduce novel concepts or ideas, however the results of the study will be useful for other scientists using the data presented or data deriving from similar measurements. For this reason I recommend this paper for publication in AMT, after some revisions as outlined below.

My main comment or criticism is about the strategy the authors adopt to deal with the Averaging Kernels (AKs). I agree that the AKs may not be a sufficiently accurate tool to evaluate the smoothing error of the retrieved WV profiles. This is due both to the fact that AKs are only a "linear" approximation of the vertical response function of the measuring system (instrument plus retrieval algorithm), and to the fact that it is generally hard to setup a covariance matrix which represents properly the variability of WV from ground to the Upper-Troposphere / Lower Stratosphere (UTLS). To show the limitations (in your test case) of the smoothing error as derived from AKs and the Rodgers (2000) approach, rather than moving the AKs analysis to the supplemental material, I would have compared, in the main paper, the actual smoothing error (obtained via intercomparisons with FPH) with the estimate of the same error obtained from AKs.

We agree that adding the comparison of smoothed FPH profiles in the main text improves the quality of the manuscript. In the revised manuscript we have included smoothed FPH profiles in Figures 5 and 6, instead of showing them in the supplemental material. Additionally, we have included in the main text results of comparisons between FTIR retrievals with both un-smoothed and smoothed FPH. We kindly refer the reviewer to the revised manuscript for

additional/modified text and figures. In particular, table 3 summarizes the finding of both comparisons.

The second general comment I have is about the sensitivity analysis presented in Sect. 5. To my opinion it would be worth to better explain why, after the analysis presented in the first part of the paper, then you start to study a quite different subject, such as the mapping of WV errors on subsequent VMR retrieval of other gases.

While the first section of the paper is focused in the retrieval/comparison of water vapor, we do not consider that section 5 is completely different. The first section considers FPH water vapor as reference and the water vapor sensitivity in the retrieval of other gases also considers FPH as reference, considering that is rare to have fully resolved coincident measurements we believe this is the right place to show both. We do agree that further details might be needed to better explain this second section. In the revised manuscript we slightly have expanded the description of this second part, mainly in the introduction.

More-over, since your measurements cover the middle-infrared, I also expect a sensitivity of the retrieved VMRs to the temperature error. This is already shown in Fig. 3 for WV. What about the error on HCN, CO, and C2H6 VMRs due to the temperature error ? Do you suggest to retrieve also the temperature profile from the same measurements or you are satisfied with the temperature profiles taken from NCEP at NDACC ?

As shown in the manuscript the importance of pre-retrieving water and use it in the retrieval of other gases is important but can be gas and site specific. The effect of using a daily NCEP temperature profile versus a more refined temporal temperature profile (or joint gas/temperature profile) can also be site/gas specific. Similar as WV, full error analysis is considered HCN, CO, and $C_2H_6$, i.e., temperature profile uncertainty is considered. In the revised manuscript we added a sentence explaining this.

Previous studies, e.g., Schneider and Hase (2008); Schneider et al. (2008), have shown that a joint retrieval of temperature profiles significantly improves the quality of $O_3$. In general, we do expect similar results for other gases. The temperature sensitivity is out of the scope of the manuscript but have added a brief description of the joint approach from previous studies in the revised manuscript.

In addition, just for reference, below there is a description of our current approach in the estimation of the error in the temperature profiles.

Errors in the temperature profile can have both systematic and random components. In our sites, we quantify these components by comparing radiosonde temperature profiles at or near each site with the daily NCEP temperature profiles. Both radiosonde and NCEP temperature profiles are interpolated onto the retrieval input grid for each site. The mean and the standard deviation of the differences between the NCEP and radiosonde temperature profiles are calculated. The mean of the difference can be as the systematic component of the error, while the standard deviation of the difference can be viewed as the random component. This is carried out for several years. Then, we include this onto the forward model parameters errors to obtain the covariance matrix

for the forward model parameters. Ideally, the same can be done for WV, however high quality and coincident measurements of WV for all sites is rare, hence the importance of the sensitivity analysis given in the second part of the manuscript.

**2 Specific Comments**
P4 L27,28: Constraining is important to select the solution which, among the possible solutions of the ill-posed inversion, is the most likely on the basis of our prior knowledge.

We edited text and included the reviewer's suggestion.

P4 Eq.1: Here it is not clear if your retrieval performs only a single or several Gauss- Newton iterations, because you don't have an iteration index in the Equation. Please, also define clearly the meaning of K. Do you compute it at each iteration ? I guess K is the Jacobian of the forward model with respect to the retrieval parameters, therefore it should be re-computed at each iteration and should show an iteration index.

Thanks for catching this up. We have modified the equation and text accordingly.

P5 L2: It would be interesting here to know which is the spectral range covered by the spectrometers used, and which is the rationale behind the selection of the listed micro-windows (e.g. minimum retrieval error ?).

Measurements at BLD and MLO follow standard measurement protocols of the InfraRed Working Group (IRWG) of NDACC. Several optical band pass filters are used to maximize the signal to noise in the middle and near infrared ($\sim 700 - 5000$ cm$^{-1}$), described briefly in section 2.1.  As mentioned in the manuscript, we do not aim to optimize a retrieval strategy of WV but rather to use a retrieval approach that we have been following in the past year. Past studies have shown that multiple micro-windows can be used. For example, Schneider et al. (2006) applied signatures between 700–1400 cm$^{-1}$; Schneider et al. (2012) used several micro-windows to retrieve water vapor isotopologues for the FTIR MUSICA retrievals. In particular for WV they used the $\sim 2800 - 2900$ cm$^{-1}$ range. Later, Barthlott et al. (2017) removed strong micro-windows used in Schneider et al. (2012) with strong absorption and added weaker lines in the 2700 cm$^{-1}$ range. An extensive number of micro-windows can be used to retrieve water. In our case, the micro-windows we use cover a similar range ($\sim 2800 - 2840$ cm$^{-1}$) and were tested to maximize the information content and minimize total error and they give consistent results across a wide range of WV columns. We added the following sentence in section 3: *"These micro-windows have been chosen to maximize the information content and minimize total error."*

P5 L8,9: I got an idea of what the authors would like to say, however I suggest to re-phrase more clearly this sentence.

After reviewing section 3, we believe the following sentence:

*"The SNR determines how much influence the spectra has in each micro-window versus the a priori, as well as to characterize the measurement error described in section 3.1"* can be removed since the SNR is primarily used in the error analysis section 3.1.

P5 L10: Which are the "relaxed covariance matrices" that induce oscillations ? Please clarify.

The following paragraph:

*"In order to prevent sporadic vertical profile oscillations due to relaxed covariance matrices we implement ad hoc diagonal elements of Sa with a maximum variability of 50% at the surface and exponentially decreasing by altitude with a inter-layer thickness correlation coefficient. A Gaussian correlation with a length of 25 km is used for the off-diagonal elements of Sa."*

has been replaced by:

*"The Sa matrix is specified at each layer as a fraction of the a priori profile, which allows for a linear scaled retrieval. We adopted a maximum variability of 50 % in the diagonal covariance and exponentially decreasing by altitude. In order to prevent sporadic vertical profile oscillations, we include a Gaussian correlation length of 25 km in the off-diagonal elements of Sa. This Sa has been optimized in order to obtain similar information content for all a priori presented in section 4.3, a requirement for efficient processing of decades of NDACC spectra."*

P5 L14: If apodization is not used, how broad is the ILS used in the forward model to emulate the instrument effect ?

The spectra are recorded at an OPD of 250 cm giving an unapodized spectral resolution of nominally 0.004 cm$^{-1}$. At this resolution all atmospheric spectral features are fully resolved in this MIR region.  We have specified this OPD in the text to be clear.

P5 L25,26: Here it is not clear how the 0.5% rms error "on the fit" maps onto the retrieved WV. Does this rms error refer to the "residuals of the fit" or directly to the retrieved WV ?

To make it clear, the description of this has been modified accordingly. The revised paragraph is:

*"We examined the effect of using more temporally refined temperature profiles. In general, the six hourly temperature profile from the ERA-I reanalysis model, produced by the European Center for Medium-Range Weather Forecasts (ECMWF) (Dee, et al., 2011), follows the daily average temperature profile shape very well for both sites. The root mean square error (rmse) between the six hourly data of ERA-I and daily average temperature is less than 0.5% using 2013 data for both BLD and MLO and the biases are less than 0.25 % for BLD and less than 0.1 % for MLO. These results suggest daily mean temperature should be adequate for retrievals but we further investigated the sensitivity of water vapor to this variability and found that water vapor agrees within 1 % if using the daily average profile. The temperature profile uncertainty is considered in the error analysis in section 3.1."*

P6 Fig.1: It would be better to show CH4 and N2O absorption contributions∂ with different line colours.

In the updated figure species are identified with the same color for the different micro-windows.

P6 L14-16: This bias is with respect to the a-priori state vector which, in turn, will probably have some bias with respect to the real profile. Note that if the bias of the retrieval was known both in sign and amplitude, then it would be possible to correct for it...

The text has been updated with the above description.

P7 L1-5: Here I would state clearly which is your retrieval vector. Do you retrieve a WV profile using the discretization mentioned in the colour scale of Fig.2a ? Do you include further fitting parameters ? (Such as atmospheric continuum, for example).

The color code scale in Fig. 2a illustrates the kernels on the retrieval grid in the troposphere but the retrieval grid does go up to 120 km. Atmospheric continuum is not considered.

P7 L7-9: Smoothing the high-vertical-resolution profiles via the averaging kernels of the coarse-vertical-resolution experiment is not mandatory, especially if you attribute a "smoothing error" to the profile differences or if you want to try characterizing the smoothing error itself. Therefore I would simply state your choice here, without trying to find a justification, which is also rather fuzzy to my view.

This description has been removed and we further improve this description in section 4. We kindly refer the reviewer to the updated text, especially in section 4.

P7 L18: Off-diagonal elements of Se may play a very important role if the spectrum is oversampled (wrt interferogram) and/or if apodization is used. Please state explicitly that, apparently, this is not your case.

In the case of very high spectral resolution spectra that fully resolve all absorption features where no apodization is used off diagonal elements are in general rendered moot. The interferogram processing may zero fill for speed but no excessive zero filling is used.

P8 L8: Please define also the symbol Kb.

Done

P8 L17-ff: In Fig.3a the error due to interfering species is also shown. Which are the considered interfering species that are not simultaneously retrieved with WV ?

The interfering species are those simultaneously retrieved. We wish to take into account the uncertainty in those lines on the retrieval of the target species. This is typically a small contributor.

P9 L1: This sentence is not clear and may be questionable. Does this mean that your AKs are not a good estimate of the vertical response function of your system (instrument plus inversion scheme) ? Why ?

The paragraph of this section has been updated, based on previous suggestions. As mentioned before the revised manuscript shows smoothed FPH profiles and quantitative comparisons of FTIR with both un-smoothed and smoothed FPH profiles. In general, the biases are lower when compared with un-smoothed profiles pointing out that FTIR AKs may not be adequate tool to use in the comparisons. This is pointed out in the updated conclusions.

P9 L7: Please note that re-gridding via interpolation is, on its own, an arbitrary smooth- ing. So, I do not fully understand why you do not want to use AKs to smooth and re- sample high-resolution profiles prior to intercomparison (as it seems you already did in the plots presented in the supplemental material).

The revised text of this section includes the comparison with smoothed profiles, as suggested in the first general comment. We kindly refer the reviewer to the updated text

P9 L12-14: As shown in Fig. 2c, the FTS retrievals have less than 3 DOFs therefore, why using so many layers for the intercomparison ? The risk is to find biases of different sign in adjacent layers.

This might be true for most gases. In fact, we do normally use total and/or partial columns depending on the number of independent pieces of information. However, a goal of this paper is to assess the ability to retrieve water vapor gradients. Hence, we decided to use these layers. Furthermore, as stated before the averaging kernels might not be a proper tool to assess sensitivity.

P10 Sect. 4.1: The underlying idea is good, however, please note that the variability evaluated here could underestimate the real WV variability, due to the constraint of the retrievals towards the a-priori state vactor. I would have estimated from measurements the variability of the spectrum vs time and would have derived the corresponding "time- mismatch" error covariance matrices relating to the individual WV profiles, using Eq. 3. I suggest to include a comment on this regard (or change approach...).

In the manuscript is already stated that the real variability might be greater because of potential lost variability during retrieval smoothing. In general, we see the stability of the spectrum by means of the signal to noise ratio (SNR), which does not change significantly, however the we see changes in WV spectral features so we see a change in amount of WV. Note that we also aim to see variability of water vapor at several altitude layers and analyzing the spectrum would not give us that information. In the revised manuscript we have removed the last two layers since we do not have enough sensitivity.

P11 Sect. 4.2: Here I did not understand if, from this analysis, you also derive an estimate of the error component to be attributed to the difference between FTS and sonde WV profiles, due to

the spatial mismatch of the measurements. I agree that it is hard to derive such an error estimate however, lacking this estimate, I do not see very much the usefulness of this section. Please explain.

As described in this section, we aim to assess the spatial mismatch between the sonde at various altitudes and the maximum sensitivity location of the FTIR. This assessment is already complex, and to our knowledge the first time applied at various altitudes. We actually believe contains great value because as mentioned at the end of this section the spatial mismatch depends on the complex convective dynamics and not only in the coincidence time interval. It is already mentioned that a thorough assessment of the spatial variability would require measurements of an extensive area simultaneously and at different altitudes. However, in the revised text we are explaining further that an error due to spatial mismatch is not derived and only an assessment of the spatial mismatch is aimed.

P13 L2,3: The second effect of a-priori WV on the solution is not clear. Did-you mean that the a-priori WV influences the solution also because it is used as initial guess for the Gauss-Newton iterations ? (This latter effect should be negligible if the retrieval converges properly).

We have re-phrased the sentence as follow:

*"The optimal estimation method is influenced by the a priori profile because it may bias the solution of equation 1."*

P16, Sect. 5: the link between this Section and the work presented in the previous Sections of the paper is non very clear. Maybe you could state at the beginning of this Section how the work you are going to present is linked with the analysis presented earlier.

The first paragraph of this section has been modified in order to make clear the connection with previous findings.

P16 L1,2: What is the meaning of "expecting WV" in this context ? Usually an optimized microwindow selection scheme tries to avoid spectral interferences from WV and re- lated isotopologues. From the second sentence, however, I understand the opposite.

*"expected"* has been replaced by *"present"* in the revised text.

P17 Table 3: Which is the rationale for the adopted sorting of interfering species in the Table ? Perhaps their relative importance ? In this case one should assess the retrieval errors due to the interference of ozone. In this spectral region I also expect temperature knowledge to be of importance (see general comments above). Moreover, I understand that the micro-windows used were selected in already publishes papers, however you could at least mention here the rationale with which they were selected / optimized (e.g. with the aim of minimizing the total retrieval error of the gas to be retrieved).

We added the following sentence in the revised text:

*"Table 3 presents the interfering species with strong and/or weak absorption signatures within each micro-window for all target gases. In all cases, the selected settings have been chosen in order to maximize the information content and minimize the total error in the retrieval."*

Furthermore, the uncertainty due to spectroscopic absorption of ozone and other species (interfering species) is considered for the final error analysis. Since this is described for water vapor, we just added the following sentence:

*"Similar as WV, full error analysis is performed, i.e., mainly considering measurement noise error and forward model parameter error (see Sect. 3.1)."*

Please see also our comment above regarding the temperature and error analysis.

**3 Technical Corrections**

P2 L26: Remove double comma between "Zugspitze" and "Germany".

Double comma removed.

P4 L3: Description... described elsewhere (rewording needed).

The sentence has been changed to:

*"A thorough description of the FPH measurement technique has been described in Hurst et al. (2011b) and Hall et al. (2016)."*

P7 L3: but as explained before, one of the goals...

Done

P11 Caption of Fig. 5: Same as Fig. 4 but for MLO.

Corrected.

P12 L9: As mentioned above, the initial spatial difference...

Corrected.

P13 L17: I do not see the usefulness of putting some figures in the supplemental material when these are recalled and described in the text of the main paper. I would put all the figures in the main paper file (of course only if this operation does not cost too much!)

As mentioned above, two of the initial supplemental figures have been removed (since the smoothed profiles have been included the main text). We decided to keep three figures in the supplement for at least two reasons: (1) we would like to provide a complete and clear process

during the analysis; (2) these images are not critical part in the main text, but of course they are mentioned in the main text so the reader can check them out if they want.

P16 Fig. 9: in the vertical axis labels "ppmv" has a small "v", is this an intentional choice ?

The subscript has been removed.

References

Barthlott, S., Schneider, M., Hase, F., Blumenstock, T., Kiel, M., Dubravica, D., García, O. E., Sepúlveda, E., Mengistu Tsidu, G., Takele Kenea, S., Grutter, M., Plaza-Medina, E. F., Stremme, W., Strong, K., Weaver, D., Palm, M., Warneke, T., Notholt, J., Mahieu, E., Servais, C., Jones, N., Griffith, D. W. T., Smale, D., and Robinson, J.: Tropospheric water vapour isotopologue data ($H^{16}O$, $H^{18}O$, and $HD^{16}O$) as obtained from NDACC/FTIR solar absorption spectra, Earth System Science Data, 9, 15–29, doi:10.5194/essd-9-15-2017.
Schneider, M. and Hase, F.: Technical note: Recipe for monitoring of total ozone with a precision of 1 DU applying mid-infrared solar absorption spectra, Atmos. Chem. Phys., 8, 63–71, 2008, http://www.atmos-chem-phys.net/8/63/2008/.

Schneider, M., Redondas, A., Hase, F., Guirado, C., Blumenstock, T., and Cuevas, E.: Comparison of ground-based Brewer and FTIR total column O3 monitoring techniques, Atmos. Chem. Phys., 8, 5535–5550, 2008.

---

## Author Comment (AC2) · 10 Jan 2019

Response to Referee #1: Matthias Schneider; Dec 2018

Black: Referee's comments
Green: Author's reply

We would like to thank Matthias Schneider for his helpful comments and suggestions. We have taken all the comments into account. In our opinion, the revised has improved thanks to suggestions provided by both reviewers.

the manuscript addresses the following topics:

(1) it describes the retrieval of water vapour profile from ground-based FTIR measurements, (2) it compares the retrieval data with frost point hygrometer sonde data, (3) it assess the impact of different WV a priori data on the retrieval of water vapour, and (4) it assess the impact of different WV a priori data on the retrieval of other trace gases where water vapour is an important interfering species.

**My general comments:**

I find (4) is a nice and valuable demonstration of the importance for using actual WV profile data in order to avoid large uncertainties in the retrievals of other trace gases. The reason is that WV is very variable and not well capturing the variability results in large retrieval errors of the other species. However, this part of the paper could be further improved by inserting references on previous work where the interference error of WV has been calculated.

We appreciate this comment. It was valuable that the reviewer included references (in the specific comments) from previous work. We have included the references in the revised manuscript. Please see also our response in the specific comment provided below.

I think (1)-(3) need revisions. A constrained remote sensing data product (here x_r) means that a priori data (here x_a) has been updated with a measurement. The product (x_r) strongly depends on the a priori data (x_a). In particularly if x_a is variable on small scales (like for WV) the variability in x_r will, to a large extent, reflect the variability of the prescribed x_a. Instead of assessing the quality of x_r the authors should assess how the remote sensing measurement can improve the assumed a priori state of the atmosphere, i.e. the authors should validate x_r-x_a by comparing it to A(x_s-x_a), where x_s is the FPH reference.

This point is well taken and along with suggestions from reviewer #2. In the revised manuscript we have included comparison of retrieved WV and sonde FPH using the formalisms by Rodgers and Connor (2003). Figures 5 and 6 include smoothed FPH profiles using equation 4 in Rodgers and Connor (2003), instead of showing them in the supplemental material (like the initial version). Additionally, we have included in the main text, results of comparisons between FTIR retrievals for both un-smoothed & smoothed FPH. We kindly refer the reviewer to the revised manuscript for additional/modified text and figures. In particular, table 3 summarizes the findings of both comparisons. Furthermore, in the revised section 4.3 we added a short description regarding the value of $x_r$-$x_a$. Please see our response in the specific comments below.

Furthermore, when using an a priori data that already captures most of the variability, the solution state should be more constrained (the S_a matrix should have much smaller entries) than when using an x_a that captures only few variability. However, judging from Sect. 3 it seems that the authors use a single S_a for constraining the different retrievals.

This has been addressed in the specific comment below.

**Specific comments:**

I have inserted my ideas/suggestions in the attached pdf version of the manuscript.

Best regards.

Comments provided by the referee are copied from the pdf and shown below in back.

Summary of Comments on amt-2018-283_MS.pdf

Page: 2
Number: 1 Author: pa5682 Subject: Cross-Out Date: 12/3/2018 8:01:36 PM

Accepted, text has been removed.

Number: 2 Author: pa5682 Subject: Inserted Text Date: 12/3/2018 7:55:01 PM
uses

Accepted.

Number: 3 Author: pa5682 Subject: Inserted Text Date: 12/3/2018 10:25:15 PM
/FTIR spectra measured at 12 different sites

Included with minor edits.

Number: 4 Author: pa5682 Subject: Inserted Text Date: 12/3/2018 10:36:31 PM
for generating a long-term data set of global representativeness of tropospheric water vapour profiles with a DOFS of almost 2.8 and of about 1.6 for the ratio between the most abundant isotopologue H216O and the heavy isotopologue HD16O.

Included with minor edits.

Number: 5 Author: pa5682 Subject: Inserted Text Date: 12/3/2018 8:03:40 PM
Comparisons of FTIR and operational radiosondes have been used to validate optimized WV profile retrieval strategies (e.g. Schneider et al. 2006; Schneider and Hase, 2009; Schneider et al., 2016).

Included with minor edits.

The complete paragraph now is:

*MUSICA (MUlti-platform remote Sensing of Isotopologues for investigating the Cycle of Atmospheric water) is a project within the NDACC/FTIR using standard spectra from a subset of NDACC sites in order to generate a long-term data set of tropospheric water vapor profiles with degrees of freedom (DOF) of about 2.8 and of about 1.6 for the ratio between the most abundant isotopologue $H_2^{16}O$ and the heavy isotopologue (Schneider:2012, 2016, Barthlott, et al., 2017). Comparisons of FTIR and operational radiosondes have been used to validate optimized WV profile retrieval strategies, (Schneider et al., 2006; Schneider and Hase, 2009). Vogelmann et al. (2015) studied the spatial-temporal variability of WV in the free troposphere (Zugspitze, Germany) by exploiting the geometry of measurements of differential absorption lidar (DIAL) and FTIR. In particular, they assessed the variability in short scales, i.e., few kilometers and minutes."*

Number: 6 Author: pa5682 Subject: Inserted Text Date: 12/3/2018 10:37:43 PM
made at two different sites, that have so far not been considered within MUSICA. We use spectral microwindows that are not identical to those of
MUSICA (Barthlott et al., 2017, Fig. 1 therein) and perform the inversion on a linear scale (instead of a logaritmic scale used by MUSICA

Thanks for pointing this out. Rather than in the introduction these details have been included in Sect 3 (Retrieval of water vapor from FTIR). The following sentence is now in Sect 3:

*"We use spectral micro-windows that are not identical to those of current MUSICA version (Barthlott et al., 2017) and perform the inversion on a linear scale (instead of a logarithmic scale used by MUSICA)."*

Page: 5
Number: 1 Author: pa5682 Subject: Highlight Date: 12/7/2018 7:19:46 PM
For constraining the authors use the same $S\_a^{-1}$ for the different a priori from Section 4.3?
If yes, the solution state will be much looser constraint for the daily varying a priori than for the monthly varying a priori, i.e the two retrievals are difficult to be compared.

In principle, we agree that Sa might need an adjustment depending on the a priori, especially for gases with less variability. The variability of water vapor can be large and not necessarily the Sa of the daily a priori is always much looser than the monthly profile. There are cases where even the daily a priori is distant from the "real" or retrieved WV but the monthly a priori is better. Optimizing a Sa for one day might not necessarily be best for another day. Furthermore, there are several reasons why we use a single Sa: (1) this work does not aim to optimize retrieval parameters but rather use a common retrieval approach that could be applicable to more sites; (2) the standard Sa used here has been optimized in all cases in order to obtain similar information content; (3) three out of four a priori are similar and we expect similar Sa and changing Sa would result in a more complex comparison.

We value this comment and in the revised manuscript we include this modified text (please note that it was modified also following the suggestion of reviewer #2:

*"The Sa matrix is specified at each layer as a fraction of the a priori profile, which allows for a linear scaled retrieval. We adopted a maximum variability of 50 % in the diagonal covariance and exponentially decreasing by altitude. In order to prevent sporadic vertical profile oscillations, we include a Gaussian correlation length of 25 km in the off-diagonal elements of Sa. This Sa has been optimized in order to obtain similar information content for all a priori presented in section 4.3, a requirement for efficient processing of decades of NDACC spectra."*

In addition, in the conclusions we mentioned that further optimization of the a priori covariance matrix might be needed in future research. The paragraph reads as:

*"Further research would explore the additional WV absorption features in order to improve the information content, e.g., micro-windows employed in the latest MUSICA version. Also, as we show, the ERA-I WV profiles yield lower biases, hence we would construct a priori covariance matrices for these that maximize accuracy and vertical structure."*

Page: 6

Number: 1 Author: pa5682 Subject: Highlight Date: 12/7/2018 6:24:45 PM
It is important to describe here the remote sensing measurement as an update of the a priori information, i.e. the actual measurement is x_r-x_a not x_r! The authors should explain how the a priori affect the retrieval results: x_r = A(x-x_a)+x_a+Dx, where Dx are retrieval errors. Because A is not an identity matrix x_r will always significantly depend on x_a.

We include a description of the effect of the apriori in Section 4.3:

*"The optimal estimation method is influenced by the a priori profile because it may bias the solution of equation 1. Since WV is highly variable, even in time scale of hours, using the most accurate a priori might improve the retrieval results. In general, the retrieval of WV can be seen as an update of the a priori information."*

The authors use variable a apriori data (see Sect. 4.3) and the variability seen in the retrieved vertical profile reflects to large extent the variability prescribed by the a priori data. Unfortunately this is not correctly considered in Sect 4 of the paper. Because the authors work with a variable a priori they should evaluate the signals in x_r-x_a=A(x-x_a), because this is the mesured signal not x_r=A(x-x_a)+x_a!. Furthermore, the authors assume that if x_s can be used as a reference for the retrieved profile x_r; howrever, actually x_s is highly resolved and absolutely calibrated reference, i.e. it is a reference for the real atmospheric profile x. So correcty the authors should compare x_r-x_a with A(x_s-x_a) in order to validate the remote sensing measurement.

In the revised section 4.3 we added a short description regarding the value of $x_r-x_a$ to evaluate signal of the measurements:

*"Additionally, the difference between WV retrievals and a priori profiles ($x_r$-$x_a$) provides further evidence in the measured signal and to some extent the variability prescribed by the a priori (Rodgers and Connor, 2003). For example, this difference is about 11 +/- 38 % using ERA-6 while for WACCM is about 29 +/- 32 % for the first layer. As we expected, from these observations it can be seen that the WACCM climatology as a priori results in greater deviations compared to ERA-6."*

As pointed out before for the comparison we follow the formalism of Rodgers and Connor (2003) in addition to the un-smoothed comparisons to assess vertical gradients and avoid averaging kernels limitations.

Page: 12
Number: 1 Author: pa5682 Subject: Highlight Date: 12/4/2018 2:10:17 PM
The retrievad data x_r are almost not sensitive to atmospheric variations above 10km, so I suggest not showing layers above 10km.

We have removed the last two layers and text has been modified accordingly.

Page: 13
Number: 1 Author: pa5682 Subject: Highlight Date: 12/4/2018 2:14:12 PM
monthly varying a apriori is used, please specify?

*"a 40 year simulation (1980-2020) of the WACCM mean profiles"*

Number: 2 Author: pa5682 Subject: Highlight Date: 12/4/2018 2:14:56 PM
daily varying a priroi?

*"daily varying (ERA-d)"*

Number: 3 Author: pa5682 Subject: Highlight Date: 12/4/2018 2:15:31 PM
6 hourly varying a priori profile?

*"6 hourly varying WV vertical profiles (00, 06, 12, and 18 UTC) obtained from ERA-I (ERA-6)"*

Number: 4 Author: pa5682 Subject: Highlight Date: 12/4/2018 2:16:28 PM
daily varying a priori profile

*"daily varying NCEP/NCAR (NCEP-d) reanalysis WV profiles"*

Number: 5 Author: pa5682 Subject: Highlight Date: 12/4/2018 2:28:19 PM
If x_s is the reference x_r-x_s=A*(x_s-x_a)-(x_s-x_a)+Dx=(A-I)*(x_s-x_a)+Dx
Dx are the retrieval errors

This is correct but we do not have anything to add.

Page: 14
Number: 1 Author: pa5682 Subject: Highlight Date: 12/4/2018 2:36:40 PM
x_r-x_s=(I-A)*(x_a-x_s)+Dx, i.e. it depends on the retrieval errors Dx and on (x_a-x_s).
Because a daily or even 6 hourly varying x_a better captures the actual variability of atmospheric
WV, using x_a from ERA-d and ERA-6 (instead of monthly climatologies) better captures the
variability as given in x_s, i.e. x_r-x_s shows a particular small scatter. This is no surprise.

In general, this might be true, but in this work, we present a quantitative assessment of the
different a priori, even for daily or 6 hourly.

Page: 15
Number: 1 Author: pa5682 Subject: Highlight Date: 12/3/2018 10:59:53 PM
see comment on Table 2

ok

Number: 2 Author: pa5682 Subject: Highlight Date: 12/3/2018 11:39:37 PM
Here retrieval data generated by using varying a priori data are compared to the sonde
measurements. This does not allow robust conclusions on the quality of the FTIR measurements.
Actually there will be a very good agreement already by comparing the varying a priori with the
sonde measurement (i.e. without any information from the FTIR measurement).
What you need to compare and evaluate is the difference with respect to the apriori! So you have
to calculate x_r-x_a and correlate it to A*(x_s- x_a).

In the revise table we have added the comparison using the formalisms by Rodgers and Connor
(2003), i.e., smoothing the FPH profiles using the water vapor averaging kernels. Note that
results using "un-smoothed" are also shown. The limitations of the averaging kernels are seen
clearly in this table.

Page: 16
Number: 1 Author: pa5682 Subject: Highlight Date: 12/5/2018 3:03:57 PM
There is some work:
The impact of interferences from WV on the retrieval of CO has been estimated by Sussmann
and Borsdorff, 2007 (doi:10.5194/ acp-7-3537-2007), on the retrieval of O3 by García et al. 2014
(doi:10.5194/amt-7-3071-2014) and on the retrieval of CH4 by Sepúlveda et al., 2014
(doi:10.5194/amt-7-2337-2014).

Thanks for providing the references of previous works. The text in the revised manuscript has
been edited to include the references provided. We note that García et al. (2014) and Sepúlveda
et al. (2014) retrieved water vapor in a first step to minimize errors in the retrieval of $O_3$, and
$CH_4$, respectively. Sussmann and Borsdorff. (2007 quantified the impact of water vapor in the
retrieval of CO and further apply a retrieval strategy to remove interference errors. Still, the
effect of using co-located and highly-resolved WV are missing in the literature.

Page: 17

Number: 1 Author: pa5682 Subject: Highlight Date: 12/7/2018 6:45:22 PM
The strategy is sufficient to avoid WV interferences in the retrievals of other trace gases and the obtained WV profiles are of a reasonable quality. However, using retrievals with more WV lines, and retrievals on log scale (tropospheric water vapour is log-normally distributed) should theoretically provide better results. Actually the MUSICA WV data show higher DOF and agreement with radiosonde also within 10-20%

The above suggestion has been included and reads as follow:

*"This example suggests that the current retrieval strategy of WV is suitable to avoid WV interference in the retrievals of other trace gases."*

In addition, in the conclusions we included the following:

*"Further research would explore the additional WV absorption features in order to improve the information content, e.g., micro-windows employed in the latest MUSICA version. Also, as we show, the ERA-I WV profiles yield lower biases, hence we would construct a priori covariance matrices for these that maximize accuracy and vertical structure."*